# A cell fate decision map reveals abundant direct neurogenesis bypassing intermediate progenitors in the human developing neocortex

Laure Coquand[1,2,8], Clarisse Brunet Avalos [1,8] ✉, Anne-Sophie Macé[3], Sarah Farcy[1], Amandine Di Cicco[1], Marusa Lampic[1], Ryszard Wimmer[1,2], Betina Bessières[4], Tania Attie-Bitach [4], Vincent Fraisier[3], Pierre Sens[5], Fabien Guimiot [6], Jean-Baptiste Brault [1] & Alexandre D. Baffet [1,7] ✉

The human neocortex has undergone strong evolutionary expansion, largely due to an increased progenitor population, the basal radial glial cells. These cells are responsible for the production of a diversity of cell types, but the successive cell fate decisions taken by individual progenitors remain unknown. Here we developed a semi-automated live/fixed correlative imaging method to map basal radial glial cell division modes in early fetal tissue and cerebral organoids. Through the live analysis of hundreds of dividing progenitors, we show that basal radial glial cells undergo abundant symmetric amplifying divisions, and frequent self-consuming direct neurogenic divisions, bypassing intermediate progenitors. These direct neurogenic divisions are more abundant in the upper part of the subventricular zone. We furthermore demonstrate asymmetric Notch activation in the self-renewing daughter cells, independently of basal fibre inheritance. Our results reveal a remarkable conservation of fate decisions in cerebral organoids, supporting their value as models of early human neurogenesis.

The human neocortex, composed of billions of neuronal and glial cells, is at the basis of higher cognitive functions[1]. Its evolutionary size expansion is particularly important in the upper layers, leading to increased surface area and folding[2]. This is largely due to progenitor cells called basal radial glial cells (bRGs), also known as outer radial glial cells[3–6]. These cells are highly abundant in humans—but rare in mice[7,8]—and reside in the outer subventricular zone (OSVZ)[9].

bRG cells derive from apical (also known as ventricular) RG cells but have lost their connection to the ventricular surface (Fig. 1a)[10,11]. A major feature of bRG cells is the presence of an elongated basal process along which newborn neurons migrate, though various morphologies have been reported including the presence of an apical process that does not reach the ventricle[12]. bRG cells undergo an unusual form of migration called mitotic somal translocation (MST), which occurs

[1]Institut Curie, PSL Research University, CNRS UMR144, Paris, France. [2]Sorbonne Université, Ecole Doctorale complexité du vivant, Paris, France. [3]UMR 144-Cell and Tissue Imaging Facility (PICT-IBiSA), CNRS-Institut Curie, Paris, France. [4]UF Embryofœtopathologie, Hopital Necker-enfants malades, Paris, France. [5]Institut Curie, PSL Research University, CNRS UMR168, Paris, France. [6]UF de Fœtopathologie – Université de Paris et Inserm UMR1141, Hôpital Robert Debré, Paris, France. [7]Institut national de la santé et de la recherche médicale, Paris, France. [8]These authors contributed equally: Laure Coquand, Clarisse Brunet Avalos. ✉e-mail: clarisse.brunet@curie.fr; alexandre.baffet@curie.fr

shortly before cytokinesis[13]. Consistent with a steady increase of the bRG cell pool during development, live imaging experiments have documented their high proliferative potential[4,5,12]. bRG cells are believed to increase the neurogenic output of the cortex while providing extra tracks for radial migration and tangential dispersion of neurons[14].

Genomic analyses have revealed the transcriptional profile of bRG in the human developing neocortex[15–17]. They have also highlighted the conservation of cellular identities between fetal tissue and cerebral organoids, despite some degree of metabolic stress[18–23]. Such studies led to the identification of several human bRG-specific genes with important roles in bRG cell generation and amplification[16,24–26]. These methods nevertheless do not allow identification of the cell fate decisions taken at the single-progenitor level that lead to this diversity[27,28]. Indeed, the sequence of progenitor divisions cannot be predicted from their final cellular output[29]. Identifying these progenitor cell fate decision modes is critical to understand how neurogenesis is regulated across species, and affected in pathological contexts. Before gliogenic stages, bRG cells can, theoretically, undergo several division modes: symmetric proliferative (two radial glial (RG) daughters), symmetric self-consuming (two differentiating daughters) or asymmetric self-renewing divisions (one RG and one differentiating daughter). Moreover, differentiating divisions can lead to the production of a neuron (direct neurogenic division) or an intermediate progenitor (IP, indirect neurogenic division).

Here, we developed a method to quantitatively map human bRG cell division modes. Using a semi-automated, live-fixed, correlative imaging approach that enables bRG daughter cell fate identification following division, in space and in time, we have established a map of cell fate decisions in human fetal tissue and cerebral organoids. We observe a remarkable similarity of division modes between the two tissues, and identify two remarkable behaviours: abundant symmetric amplifying divisions, as well as frequent self-consuming direct neurogenic divisions, suggesting an alternative route to the asymmetric self-renewing divisions that dominate in mouse apical radial glial progenitor (aRG) cells. Within these asymmetrically dividing cells, we demonstrate asymmetric Notch signalling, independent of basal process inheritance.

## Results

### Morphological identification of bRG cells

First, we validated the identification of bRG cells based on morphological features. Human fetal pre-frontal cortex tissues from gestational weeks (GW) 14–18 were stained for phospho-Vimentin (p-VIM), which marks mitotic RG cells. Imaging within the SVZ revealed four different morphologies for these cells: unipolar with a single apical process, unipolar with a basal process, bipolar with both an apical and a basal process, and cells with no visible process (Fig. 1b,c). Bipolar bRG cells always had a major thick process and a minor thin process, which could be apical or basal (Fig. 1b, 2P). Overall, over 80% of p-VIM+ cells displayed at least one process, and 60% a basal process. All process-harbouring p-VIM+ cells were also SOX2+, while 20% of non-polarized p-VIM+ cells were negative for SOX2 (Fig. 1d).

We then explored bRG cell morphology in non-mitotic cells. We first validated bRG identity by showing that they were positive for HOPX, PTPRZ1, LIFR and SOX2 (bRG cells)[15], but negative for EOMES (IPs), and HuC/D and NEUN (neurons) (Extended Data Fig. 1a–d). We note that a small subset of SOX2+/EOMES−/NEUN− may be oligodendrocyte progenitor cells (OPCs), which begin to appear around these stages. Fetal brain slices were infected with GFP-expressing retroviruses (RVs) and stained for SOX2, EOMES and NEUN (Fig. 1e and Extended Data Fig. 1e,f). This analysis confirmed that over 80% of SOX2+/EOMES−/NEUN− cells displayed apical and/or basal processes, while 20% were non-polarized (Fig. 1f,g). Moreover, the majority of process-harbouring cells were SOX2+/EOMES−/NEUN−, and around 40% of non-polarized cells were SOX2+/EOMES−/NEUN− (Fig. 1h). Therefore, human fetal bRG cells largely display elongated processes, though 20% are non-polarized.

Next, we performed live imaging of GFP-expressing cells in fetal slices, focusing on elongated bRG cells. Dividing cells had the same morphology as previously described in fixed samples (Extended Data Fig. 1g). The majority of process-harbouring cells performed MST, though 25% performed stationary divisions (Fig. 1i,j and Supplementary Video 1). MST could occur in the apical direction or the basal direction, depending on their shape. When bRG cells had two processes, MST occurred in the dominant (thick) process (Fig. 1j).

Finally, week 8–10 cortical organoids were infected with RV and stained for the cell fate marker SOX2, EOMES and NEUN, which revealed abundant SOX2+ bRG cells above the ventricular zone (Fig. 1k and Extended Data Fig. 1h,i). As in fetal tissue, the majority of SOX2+/EOMES−/NEUN− cells displayed one or two elongated processes, and 20% were non-polarized (Fig. 1l,m). The vast majority of process-harbouring cells, and around 40% of non-polarized cells, were SOX2+/EOMES−/NEUN− (Fig. 1n). Live imaging confirmed these morphologies and indicated that the majority of bRG cells performed MST (Fig. 1o and Extended Data Fig. 1j). Therefore, the majority of human bRG cells can be identified in live samples based on their elongated morphology and ability to divide, which is conserved between fetal tissue and organoids.

### A correlative imaging method to identify cell fate decisions

Next, we developed a method to identify the fate acquired by daughter cells following progenitor cell division in cerebral organoids. We established a correlative imaging method consisting of live imaging GFP-expressing progenitors and, following fixation and immunostaining, assigning a fate to the live-imaged cells (Fig. 2a). We developed a computer-assisted method to automate the localization of the videos in the immunofluorescence images (Supplementary Information). In brief, RV-infected tissue slices are live imaged for 48 hours and, at the end, 4X brightfield images of the slices containing positional information from each video are generated (Fig. 2b and Supplementary Video 2). Slices are then fixed, stained for the cell fate markers SOX2, EOMES and NEUN, and tile scan images of the entire slices are acquired. Both live and fixed images are automatically segmented, paired, flipped and aligned. The position of each video is thereby obtained on the immunostained images, leading to the identification

---

**Fig. 1 | Morphological characterization of bRG cells in human cerebral organoids and fetal tissue. a**, Schematic representation of human neocortex development. CP, cortical plate. **b**, Phospho-Vimentin immunostaining of human frontal cortex at GW 18. Image is overexposed to visualize processes, revealing cells with basal process (bP), apical process (aP) and both processes (2P). **c**, Quantification of mitotic bRG cell morphologies in GW 14–18 frontal cortex. N = 3 brains, 338 cells. **d**, Proportion of p-VIM+ cells positive for SOX2, depending on morphology. N = 3 brains, 456 cells. **e**, SOX2, EOMES and NEUN immunostaining in human frontal cortex at GW 17. **f**, Morphologies of GFP-expressing SOX2+ cells in human frontal cortex at GW 17. **g**, Quantification of morphologies of GFP-expressing SOX2+ cells in human frontal cortex at GW 14–17. N = 2 brains, 350 cells. **h**, Proportion of SOX2+/EOMES− and EOMES+ (with or without SOX2) progenitors, depending on morphology in human frontal cortex at GW 14–17. N = 2 brains,

204 cells. **i**, Live imaging of bRG cell performing MST in human fetal tissue. The arrowhead indicates basal process. **j**, Directionality of MST depending on bRG cell morphology in human frontal cortex at GW 14–18. N = 3 brains, 242 cells. **k**, SOX2, EOMES and NEUN immunostaining in week 8 cerebral organoids (top). Schematic representation of week 8–10 cerebral organoids (bottom). **l**, Morphologies of GFP-expressing SOX2+ cells in cerebral organoids at weeks 7–10. **m**, Quantification of morphologies of GFP-expressing SOX2+ cells in cerebral organoids at weeks 7–8. N = 2 batches, 104 cells. **n**, Proportion of SOX2+/EOMES− and EOMES+ (with or without SOX2) progenitors, depending on morphology in cerebral organoids at weeks 8–10. N = 3 batches, 205 cells. **o**, Directionality of MST depending on bRG cell morphology cerebral organoids at weeks 8–9. N = 4 batches, 260 cells. Arrowheads indicate bRG processes and asterisks indicate soma. Data are presented as mean values +/− s.d.

of matching cells between the live and fixed samples (Fig. 2b). Using this method, dividing bRG cells can be live imaged and the fate of the two daughter cells identified (Fig. 2c and Supplementary Video 3).

Daughter cell fate was analysed on average 30 hours after division. We noted that when a daughter cell differentiated (for example, into an EOMES+ IP), it often retained some expression of the mother

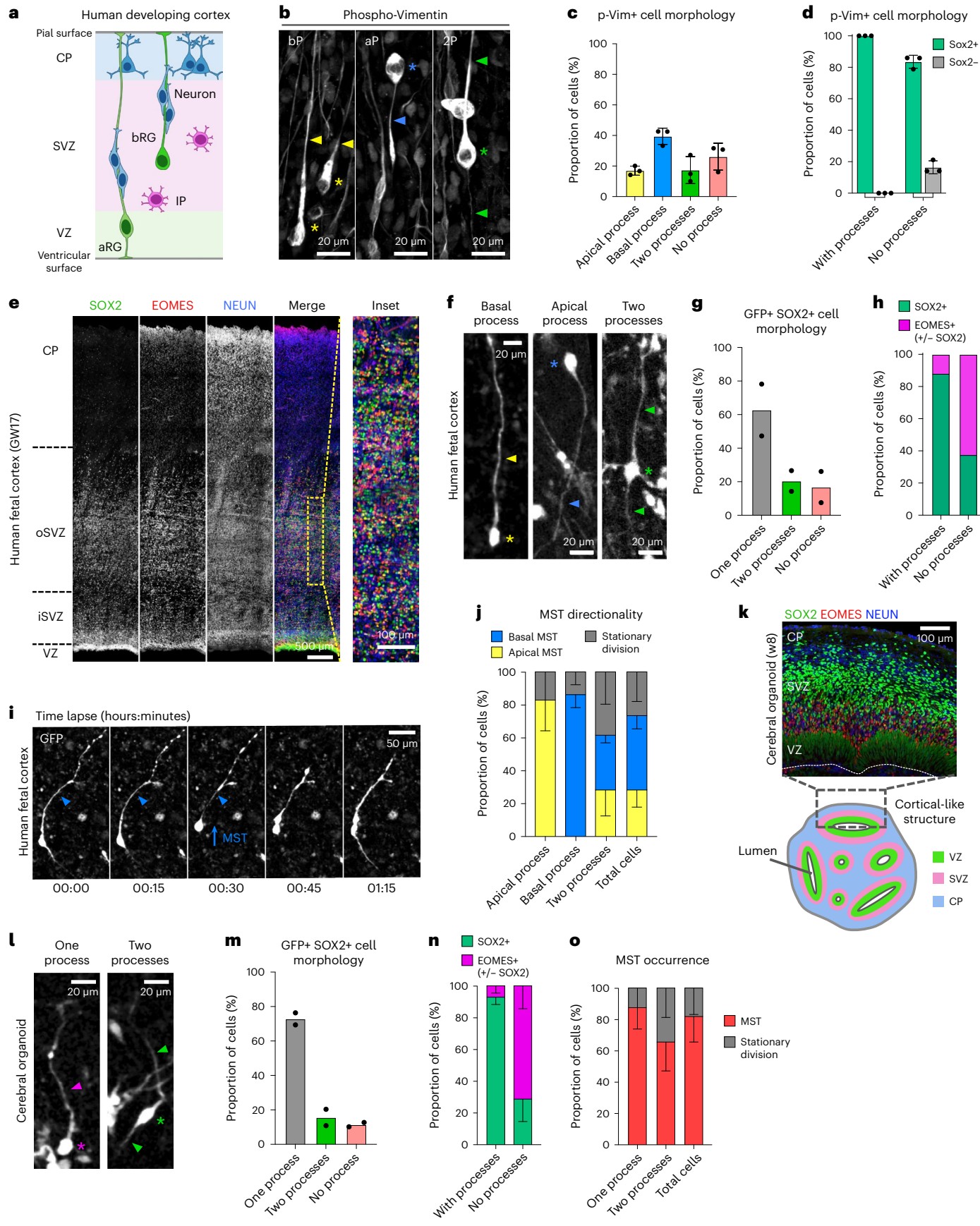

cell fate marker (SOX2), irrespective of the division mode (Fig. 2c). Expression of a novel fate marker was on the contrary very rapid, with EOMES or NEUN being detected in daughter cells that had divided 1–2 hours before the end of the movie (Extended Data Fig. 2a). Moreover, putative IPs and migrating neurons could be live imaged and cell fate analysed at the last timepoint (Fig. 2d,e and Supplementary Videos 4 and 5). Therefore, this semi-automated correlative microscopy method allows the identification of cell fate markers in live-imaged cerebral organoids, in a highly reproducible and quantitative manner.

### A map of cell fate decisions in cerebral organoids

To generate a map of progenitor division modes, we analysed 1,101 dividing bRG cells, in weeks 7 to 9 and 13 to 15 cerebral organoids, prior to the start of gliogenesis[18,20] (Fig. 3a, Extended Data Fig. 2b,c and Supplementary Videos 6–8). We report the relative probabilities, through time, of all possible division modes (Fig. 3b for week 8, and Extended Data Fig. 3 for all other stages). Notably, we never observe asymmetrically dividing bRG cells generating one IP and one neuron. We first quantified the fraction of proliferative (amplifying) divisions (leading to two SOX2+ cells) versus neurogenic divisions (leading to at least one differentiating cell, EOMES+ or NEUN+). This analysis revealed a high rate of bRG cell amplification, which increases between weeks 7 and 9, and decreases between weeks 13 and 15 (Fig. 3c). Within neurogenic divisions, bRG cells performed symmetric self-consuming divisions, leading to two differentiating cells, or asymmetric self-renewing divisions, leading to one bRG cell and one differentiating cell. Self-consuming divisions decreased between weeks 7 and 9 and increased between weeks 13 and 15 (Fig. 3d). In both types of neurogenic divisions (asymmetric or symmetric), bRG cells could divide directly into neurons or indirectly, via the generation of IPs. Strikingly, we observed that direct neurogenic divisions dominated in human bRG cells, indicating that the generation of IPs is not a systematic differentiation trajectory in these cells (Fig. 3e). These divisions decreased between weeks 7 and 9 and increased between weeks 13 and 15 (Fig. 3e).

Next, we modelled how these different modes of progenitor divisions affected their final output (Methods). At each stage, we predicted the average number of bRG cells, IPs and neurons generated from a single bRG cell, after four rounds of division, which corresponds to approximately 1 week of development (Fig. 3f for week 8 and Extended Data Fig. 4a for all other stages). At week 8, a single bRG cell leads on average to the generation of 5.75 bRG cells, 1.21 IPs and 2.69 neurons, highlighting their strong self-amplification potential (Fig. 3f). Modelling bRG output through time reveals that bRG amplification increases from week 7 to 9 and decreases from 13 to 15 (Fig. 3g). Strikingly, this occurs at a relatively constant neurogenic rate indicating that, at the single-progenitor level, self-amplification varies but not the number of differentiated cells produced. Finally, we tested how variations in cell fate decision probabilities would affect their output. Reducing the rate of proliferative divisions by 20% in favour of asymmetric self-renewing indirect divisions (one bRG and one IP)—the dominant division mode in mouse aRG cells at neurogenesis onset—reduced the total production of bRG cells by 31% after only four divisions (Extended Data Fig. 4b). Overall, this analysis indicates that bRG cells are highly proliferative and undergo important self-amplification. Upon differentiation, they undergo frequent self-consuming terminal divisions, as well as abundant direct neurogenesis.

### A map of cell fate decisions in human fetal tissue

Next, we adapted this correlative imaging method to human frontal cortex samples at GW 14–18. Although slices were substantially larger, the macro proved to be equally efficient at automatically identifying and aligning corresponding regions between the live and fixed datasets, indicating that it can be used for any type of tissues (Extended Data Fig. 5a,b). We analysed the division modes of 335 human fetal bRG cells, following 48-hour live imaging (Fig. 4a, Extended Data Fig. 5c and Supplementary Videos 9 and 10). We confirmed the rapid expression of differentiation markers following cell division (Extended Data Fig. 5d). As in cerebral organoids, the majority of bRG cells performed symmetric proliferative division, generating two SOX2+ daughters (Fig. 4b,c and Extended Data Fig. 5e). At GW 18, we noted a decrease in neurogenic division in favour of gliogenic divisions, indicating that the switch begins around this developmental time. Within neurogenic divisions, we again observed abundant symmetric self-consuming divisions that remained relatively constant (around 32% of all neurogenic divisions; Fig. 4b,d Extended Data Fig. 5e). Finally, we confirmed that direct neurogenic divisions are an abundant bRG cell division mode, which again remained stable from GW 14 to 18 (over 40% of all neurogenic divisions; Fig. 4b,e Extended Data Fig. 5e). We tested whether these cell fate decisions varied depending on bRG cell mitotic behaviours (apical MST, basal MST or static division) but found no clear effect of this parameter (Extended Data Fig. 5f). Overall, we find a strong conservation of division modes between human fetal tissue and cerebral organoids, with the coexistence of asymmetric self-renewing progenitors—as classically observed in mouse aRG cells—together with self-amplifying and self-consuming neurogenic progenitors, which represents an alternative route for neuronal generation.

### Increased direct neurogenesis in the basal OSVZ

The human OSVZ is extremely large (approximately 3 mm at GW 17) and bRG cells may, therefore, be exposed to different microenvironments depending on their position, which may influence their division modes. Moreover, bRG cells progressively migrate through the SVZ and have a different history depending on their position. We, therefore, explored whether bRG division modes vary along the apico-basal axis in the human fetal brain. To test this, we adapted the above-described method to automatically record the position of each dividing bRG cell within the tissue. The position of bRG cells along the apico-basal axis only had a minor effect on symmetric proliferative versus neurogenic division (Fig. 4f,g). Similarly, the rate of symmetric self-consuming versus asymmetric self-renewing divisions was not significantly different (Fig. 4f,g). However, we observed a clear difference in the rates of direct versus indirect neurogenesis, depending on the position in the tissue. Indeed, indirect neurogenic divisions (EOMES+ cells) occurred on average 800 µm from the apical surface, while direct neurogenic divisions (NEUN+ cells) occurred much more basally, 1,306 µm from the apical surface (Fig. 4f,g). These experiments could not be performed in cerebral organoids, as they display a much smaller OSVZ. Overall, they demonstrate that dividing bRG cells undergo more direct neurogenic divisions when located in the basal part of the fetal OSVZ.

---

**Fig. 2 | A semi-automated correlative imaging method to identify cell fate decisions in cerebral organoids. a**, Schematic representation of correlative microscopy pipeline. **b**, Step-by-step protocol for semi-automated correlative microscopy. (1) bRG cells are live imaged at 20X for 48 hours. (2) 4X brightfield images containing the video coordinates are assembled. (3) Organoid slices are fixed, immunostained for SOX2, EOMES and NEUN, and imaged. (4) Images are automatically segmented to outline slices from live and fixed samples. (5) Slice contours are automatically paired based on shape and area and (6) aligned (including a horizontal flip if needed). (7) Video fields of view are automatically annotated on the immunostaining images. (8) Regions of interest are re-imaged at higher resolution (×40) and cells from live and fixed samples are manually matched. **c**, Live/fixed correlative analysis of a dividing bRG cell generating a self-renewing bRG daughter and a differentiating IP daughter. **d**, Live/fixed correlative analysis of a dividing IP cell generating two neuronal daughters. **e**, Live/fixed correlative analysis of a migrating neuron. All images are representative examples of experiments performed at least three times independently (N = 1,101 bRG cells).

## Basal-process-independent self-renewal

The mechanism of bRG cell asymmetric division remains unknown. In mouse aRG cells, growing evidence supports the role of basal process inheritance in stem cell fate maintenance[30–32]. We, therefore, used our correlative imaging method to test whether process inheritance correlates with bRG fate maintenance upon asymmetric division of

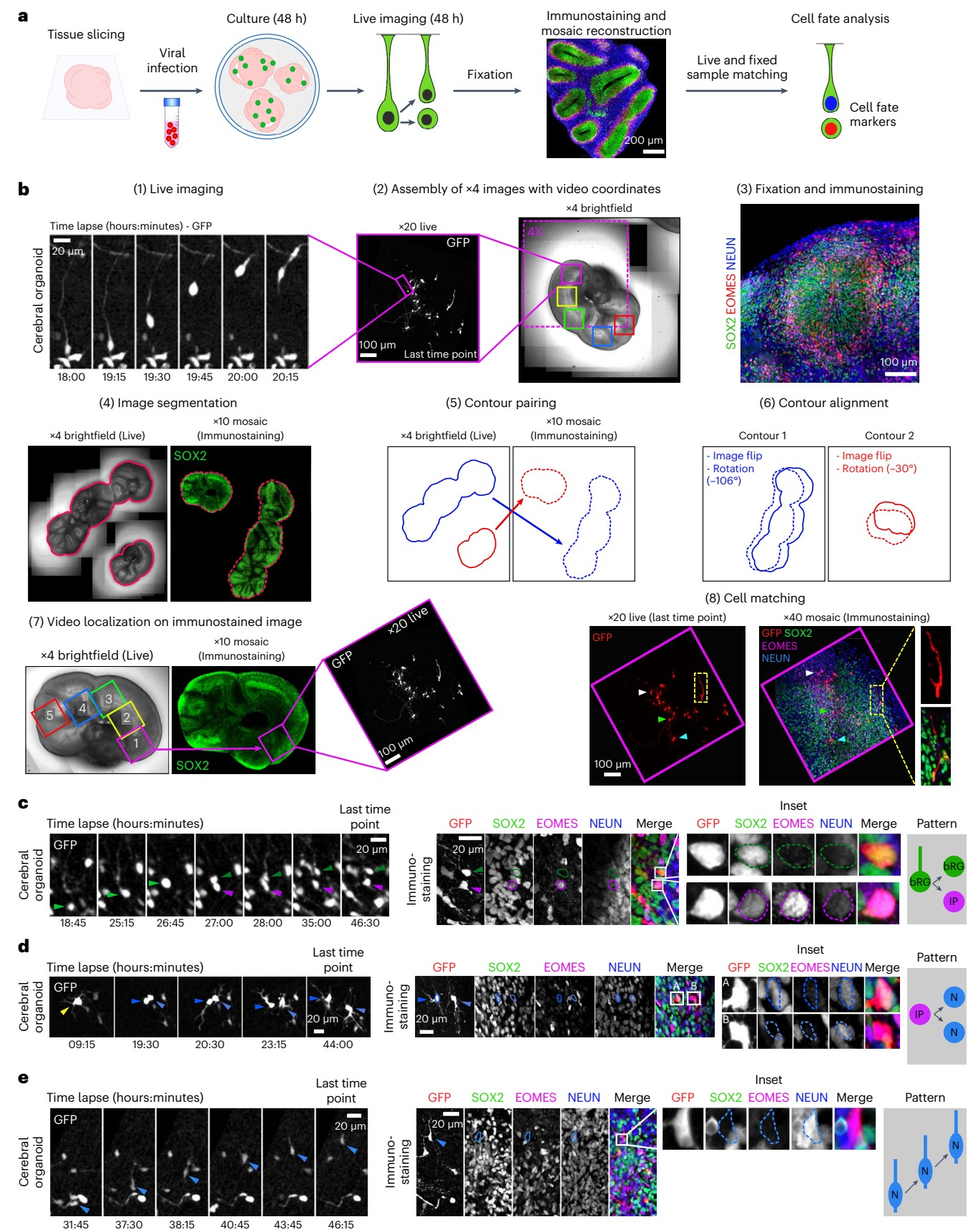

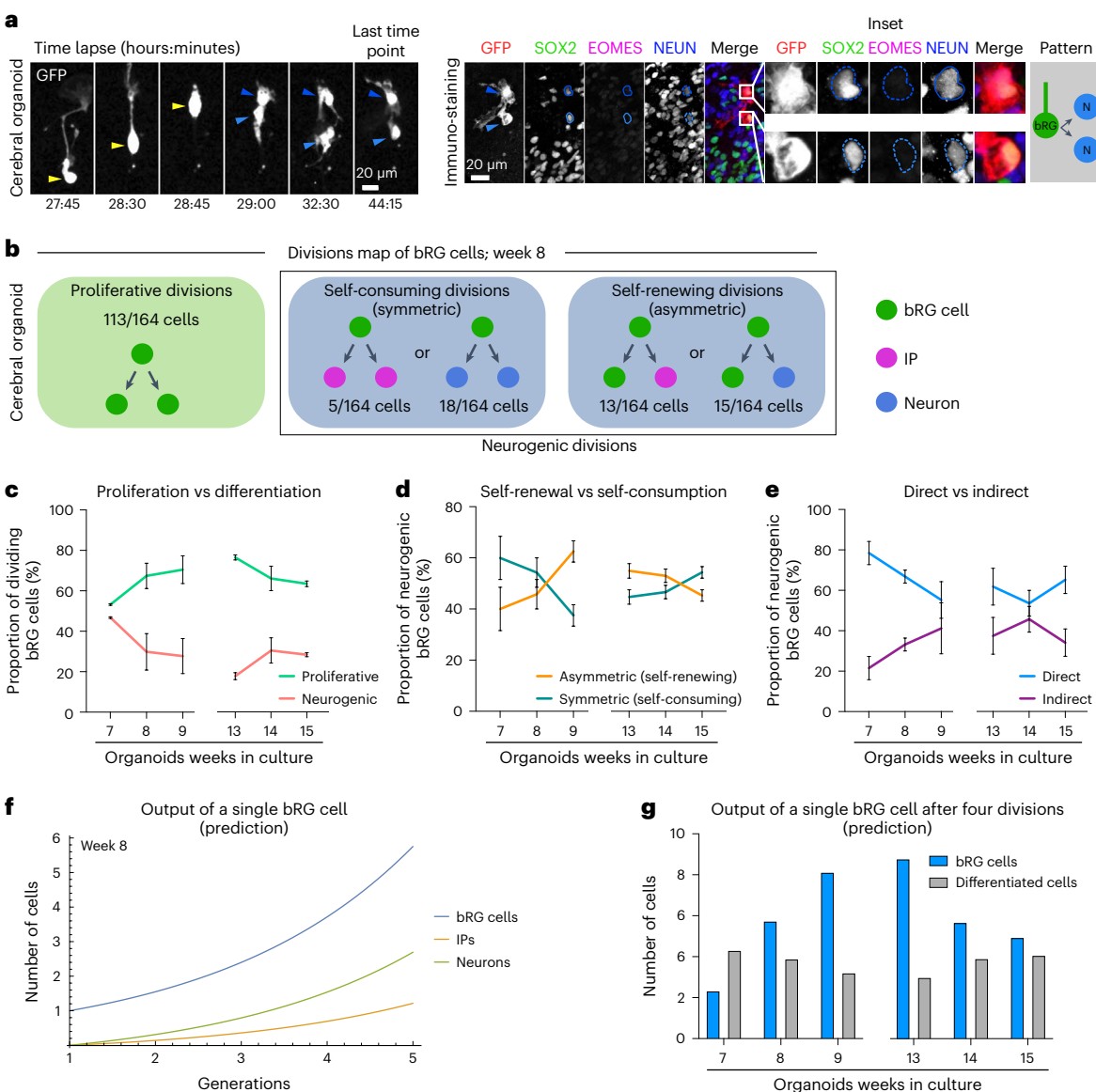

**Fig. 3 | A map of cell fate decisions in human cerebral organoids. a**, Live/fixed correlative analysis of a dividing bRG cell generating two neuronal daughters. **b**, Summary of all division patterns identified in bRG cells in week 8 cerebral organoids (*N* = 164 bRG cells). **c**, Proportion of proliferative versus neurogenic divisions of bRG cells in week 7–9 and 13–15 cerebral organoids. **d**, Proportion of asymmetric (self-renewing) versus symmetric (self-consuming) neurogenic divisions of bRG cells in week 7–9 and 13–15 cerebral organoids. **e**, Proportion of direct versus indirect neurogenic divisions of bRG cells in week 7–9 and 13–15 cerebral organoids. **c**–**e**, Week 7, *N* = 114 bRG cells and two independent live-imaged slices; week 8, *N* = 164 bRG cells and two independent live-imaged

slices; week 9, *N* = 106 bRG cells and two independent live-imaged slices; week 13, *N* = 206 bRG cells and two independent live-imaged slices; week 14, *N* = 254 bRG cells and three independent live-imaged slices; week 15, *N* = 257 bRG cells and two independent live-imaged slices. **f**, Simulation of the output of a single bRG cell after one to five generations, based on week 8 fate decision probabilities. **g**, Simulation of the output of a single bRG cell after four divisions (five generations) in week 7–9 and 13–15 cerebral organoids. All images are representative examples of experiments performed in at least two independent fetal brains. Data are presented as mean +/− s.d.

human bRG cells. We first live imaged 79 asymmetrically dividing bRG cells (one bRG daughter and one differentiating daughter) within week 8–10 cerebral organoids, and analysed daughter cell fate depending on process inheritance (Fig. 5a,b and Supplementary Videos 11 and 12). In half of these cells, process-inheriting daughters maintained a bRG fate but in the other half, process-inheriting daughters differentiated (Fig. 5c,d). This was the case whether the asymmetric divisions generated an IP or a neuron directly. We next performed a similar analysis in GW 14–17 human fetal brain tissue. We analysed 82 asymmetrically dividing bRG cells and again found no correlation between basal process inheritance and bRG cell fate (Fig. 5e,f and Supplementary Videos 13 and 14): 52.4% of basal-process-inheriting daughters remained bRG

cells, and 47.6% differentiated (Fig. 5g,h). In support of these results, SOX2+ daughter cells that did not inherit a process could be observed to regrow one after division (Extended Data Fig. 6 and Supplementary Video 15). Therefore, in human bRG cells, the basal process appears to be a consequence, rather than a cause, of bRG cell fate upon asymmetric division. Its presence during interphase may however participate in long-term bRG fate maintenance.

### Basal-process-independent Notch activation

In aRG cells, it was proposed that the basal process acts as an antenna for the reception of Notch signalling from the surrounding cells, in particular neurons[31,32]. First, we validated the role of Notch signalling

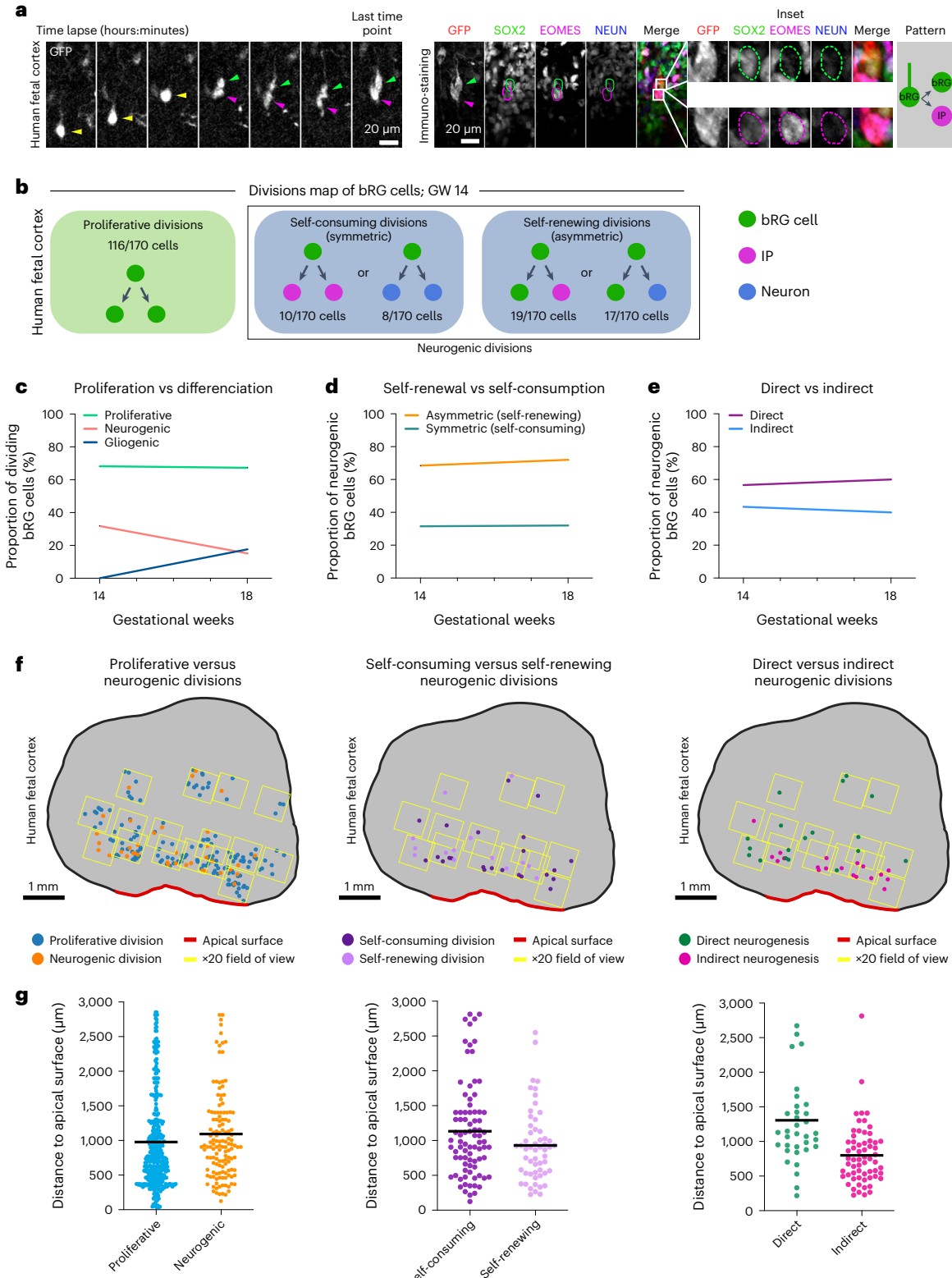

**Fig. 4 | A map of cell fate decisions in fetal human frontal cortex. a**, Live/fixed correlative analysis of a dividing bRG cell generating a bRG daughter and an IP daughter. **b**, Summary of all division patterns identified in bRG cells at GW 14 (*N* = 170 bRG cells) human frontal cortex. **c**, Proportion of proliferative versus neurogenic divisions of bRG cells in GW 14 and 18 human fetal tissue. **d**, Proportion of asymmetric (self-renewing) versus symmetric (self-consuming) neurogenic divisions of bRG cells in GW 14 and 18 human fetal tissue. **e**, Proportion of direct versus indirect neurogenic divisions of bRG cells in

GW 14 and 18 human fetal tissue. **f**, Spatial distribution of proliferative versus neurogenic (left), self-consuming versus asymmetric self-renewing (middle) and direct versus indirect neurogenic divisions (right) in GW 17 human frontal cortex. **g**, Quantification of proliferative versus neurogenic (left), self-consuming versus asymmetric self-renewing (middle) and direct versus indirect neurogenic divisions (right) in GW 17–18 human frontal cortex (*N* = 355 cells for GW 17 and 172 cells for GW 18). Data are presented as mean values. Images are representative examples of experiments performed in at least two independent fetal brains.

for bRG cell self-renewal in cerebral organoids[4]. Following retroviral infection to deliver GFP, slices were treated with the γ-secretase inhibitor DAPT—which blocks Notch signalling—for 2 days. Quantification revealed a depletion of GFP+ bRG cells and a corresponding increase of IPs (Extended Data Fig. 6a,b). The neuronal population was not affected in the timeframe of the experiment, indicating that indirect neurogenesis is the default differentiation pathway in the absence of Notch signalling. Next, we monitored Notch signalling in bRG daughter cells, depending on process inheritance. As a readout, we analysed the expression of its downstream target HES1. In cerebral organoids, HES1 was strongly expressed in the ventricular zone where aRG cells are highly abundant and in a sparse manner in the SVZ, reflecting the SOX2+ bRG cell distribution (Fig. 6a). Week 8–11 organoid slices were live imaged for 48 hours, stained for HES1, EOMES and NEUN, and processed through the correlative imaging protocol. Out of 276 bRG cell, 186 symmetric proliferative divisions, 53 asymmetric divisions and 37 symmetric self-consuming divisions (Fig. 6b). Consistent with its oscillatory behaviour in RG cells[33], HES1 was only detected in a subset of bRG cells, whether these cells were generated following symmetric or asymmetric divisions (Fig. 6c,d and Supplementary Video 16). As expected, HES1 was never detected in differentiating cells ($N$ = 90 cells; Fig. 6d). Out of 276 live-imaged bRG cells, we identified 16 cells that divided asymmetrically, with detectable HES1 expression in daughter cells (Fig. 6d). HES1 was always detected in the non-differentiating daughter (EOMES− and NEUN−), supporting preferential Notch signalling in the self-renewing bRG daughter upon asymmetric division (Fig. 6d). However, we found no correlation between HES1 expression and process inheritance: 8 HES1-expressing cells inherited the basal process and 8 did not (Fig. 6e,f and Supplementary Video 17). These data further indicate that process inheritance does not correlate with bRG cell fate, and that the basal process is not involved in differential Notch signalling upon asymmetric division in bRG cells, as it is believed to be in aRG cells.

## Discussion

bRG cells are key actors in the evolutionary expansion of the human brain, but the sequence of events leading to their massive neuronal output is unknown. Using live/fixed correlative imaging, we provide a map of their division modes at early—mostly pre-gliogenic—stages. In mice, aRG cells undergo a precise switch in division modes at E12.5, from mostly symmetric amplifying divisions to mostly asymmetric divisions generating one self-renewing aRG cell and one IP that will divide once to generate two neurons[34–36]. Here we show that, at neurogenesis onset, multiple bRG cell division modes co-exist, pointing to more complex regulation in the human cortex.

We observe that bRG amplification through symmetric cell divisions is dominant, and in organoids increases from weeks 7 to 9 and decreases from 13 to 15. Modelling reveals that this occurs at a constant rate of neurogenesis, indicating that, at each developmental stage, single bRG cells produce an equal number of differentiated cells, irrespective of their self-amplification level. At the population level, however, the gradual increase in the total number of bRG cells during

development will lead to an increase in the production of differentiated cells. These results suggest that neuronal production increases through development as a consequence of the expanding pool of bRG cells, but not of increased neurogenic potential of single bRG cells.

Our results indicate that, on top of asymmetric self-renewing divisions, bRG cells undergo symmetric amplifying divisions and self-consuming divisions, pointing to an alternative route for neuronal generation. Neurogenic divisions are frequently direct, bypassing IP production. This represents another major difference with mouse aRG cells that largely rely on IPs to amplify the neurogenic output. The evolution of cortical neurogenesis in amniotes is regulated by the balance between direct and indirect neurogenesis[37]. aRG cells in sauropsids undergo direct neurogenesis, while mammals largely rely on indirect divisions in the evolutionary more recent neocortex, a process associated with size expansion and regulated by Robo signalling levels[37]. We show that this rule does not, however, apply to bRG cells, in which direct neurogenesis is common. aRG cells may rely on IPs to amplify their neurogenic output, because their own self-amplification is limited by spatial constraints. They must indeed divide at the ventricular surface to precisely segregate their apical junctions between daughters and maintain a proper neuroepithelial structure. Interkinetic nuclear migration (INM) leads to the formation of a pseudostratified epithelium that allows an increase in the aRG cell pool, but their amplification still reaches a physical limit[38–40]. bRG cells on the other hand are not subject to this physical limitation and can amplify their own pool both radially and tangentially. In this regard, IPs may be less relied upon to increase the neurogenic output. Whether direct and indirect divisions ultimately lead to the formation of different neuronal subtypes, as observed in aRG cells, remains to be tested[37].

Cerebral organoids have emerged as a powerful system to investigate human brain development[41–43]. To what degree they faithfully recapitulate fetal neurogenesis is, however, important to monitor. Genomics studies have highlighted the similarity of transcriptional profiles, though substantial metabolic stress has been reported in organoids[18–23]. Here, we report a high similarity of bRG cell division modes between organoids and fetal tissue. We note that direct neurogenesis is slightly more abundant in organoids, which may reflect cell stage differences or inherent limitations of the organoid model. Nevertheless, an advantage of imaging approaches, such as ours, is that the organoid necrotic core can be avoided, focusing on the cortical-like lobes at the periphery of the organoids. These cortical-like structures are, however, much thinner than in the fetal brain, limiting the ability to probe how bRG cell position impacts their division modes, as performed here in fetal tissue.

The molecular mechanism regulating asymmetrical division in RG cells has been a matter of controversy. In aRG cells, increasing evidence supports a role for the basal process in cell fate, which correlates with Notch activation and self-renewal[30–32]. However, we do not observe such a correlation in human bRG cells where Notch signalling is activated in the self-renewing daughter irrespective of basal process inheritance. aRG somas are located in the ventricular zone and their basal process extends through the cortex, contacting neurons from which Notch–Delta

**Fig. 5 | Basal process inheritance does not predict bRG fate on asymmetric division. a**, Live/fixed correlative analysis of basal process inheritance in a dividing bRG cell generating a process-inheriting bRG daughter and neuron, within a cerebral organoid. **b**, Live/fixed correlative analysis of basal process inheritance in a dividing bRG cell generating a process-inheriting IP daughter and a bRG daughter, within a cerebral organoid. **c**, Distribution of cell fates depending on process inheritance upon asymmetric cell division in week 8–10 cerebral organoids ($N$ = 79 asymmetrically dividing cells from five experiments). **d**, Proportion of self-renewing versus differentiating daughter cells upon asymmetric division, depending on process inheritance in week 8–10 cerebral organoids ($N$ = 79 asymmetrically dividing cells from five experiments). **e**, Live/fixed correlative analysis of basal process inheritance in a dividing bRG cell

generating a process-inheriting bRG daughter and a neuron, within fetal frontal cortex. **f**, Live/fixed correlative analysis of basal process inheritance in a dividing bRG cell generating a process-inheriting IP daughter and a bRG daughter, within fetal frontal cortex. **g**, Distribution of cell fates depending on process inheritance upon asymmetric cell division in GW 14–17 human frontal cortex ($N$ = 82 asymmetrically dividing cells from two experiments). **h**, Proportion of self-renewing versus differentiating daughter cells upon asymmetric division, depending on process inheritance in GW 14–17 human frontal cortex ($N$ = 82 asymmetrically dividing cells from two experiments). All images are representative examples of experiments performed in at least three independent organoid batches and two independent fetal brains. N, neuron; D, differentiating daughter.

signalling can be activated. bRG somas on the other hand are located in the SVZ and both their daughter cells are in close proximity to neurons. Therefore, owing to the bRG cell microenvironment, it is consistent that

their basal process does not confer differential Notch signalling. Other factors, such as centriole age, mitochondrial dynamics, mitotic spindle positioning or Sara endosomes are promising candidates[44–46].

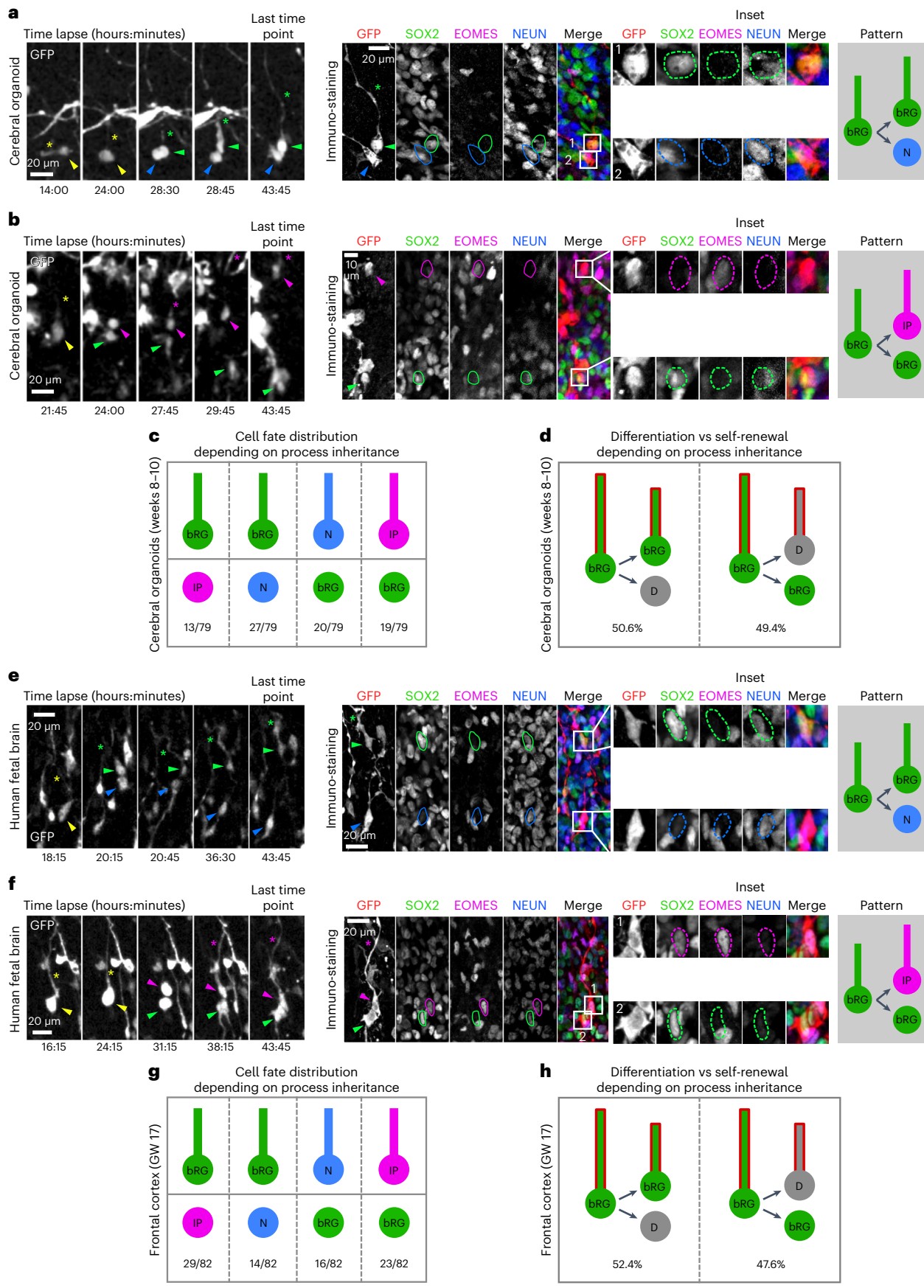

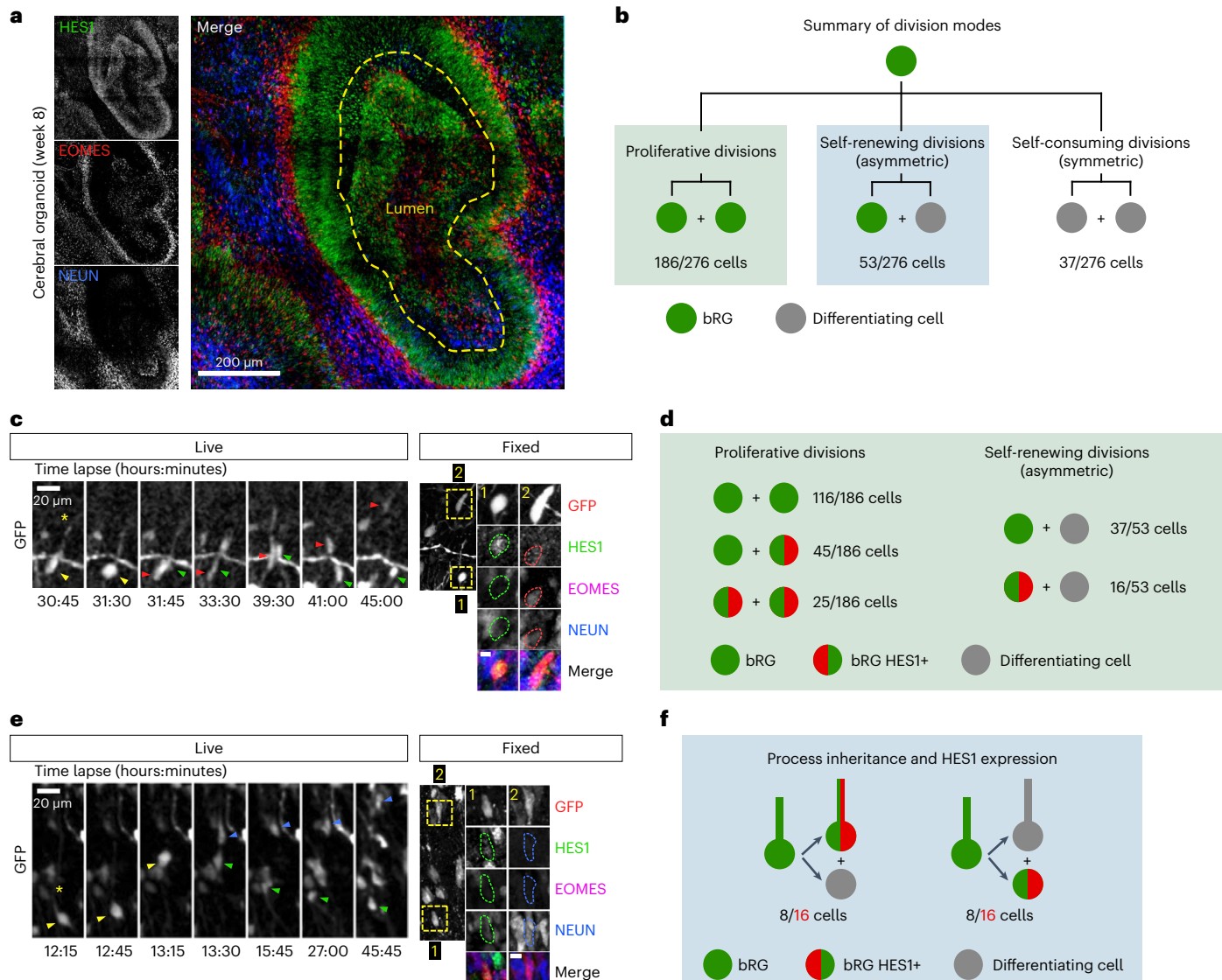

**Fig. 6 | HES1 is preferentially expressed in bRG daughters, irrespective of process inheritance. a**, HES1, EOMES and NEUN immunostaining in human cerebral organoid at week 8. **b**, Distribution of division modes identified in bRG cells within week 8–11 cerebral organoids. bRG daughter (EOMES− and NEUN−), differentiating daughter (EOMES+ or NEUN+) (N = 276 bRG cells from three batches of organoids). **c**, Live/fixed correlative analysis of an asymmetrically dividing bRG cell revealing HES1 expression specifically in self-renewing daughter (EOMES− and NEUN−). **d**, Summary of HES1 expression in daughter cells depending on division modes (N = 239 cells from three batches of organoids). **e**, Live/fixed correlative analysis in asymmetrically dividing bRG cells revealing lack of correlation between HES1 expression and basal process inheritance. **f**, Summary of HES1 expression depending on process inheritance in asymmetrically dividing bRG cells, within week 8–11 cerebral organoids (N = 16 cells from three batches of organoids). All images are representative examples of experiments performed in at least three independent organoid batches.

Descriptions of clonal relationships are a powerful means to understand cellular diversity. Key to this is the identification of the cell fate decision branch points along lineages. The semi-automated correlative imaging method enables quantitative measurement of progenitor cell division modes in human cortical tissue. This will allow to probe neuronal subtype generation or the switch to glio-genesis, through time and space, across species and in pathological contexts.

## Online content

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

## Methods

### Statistics and reproducibility
Quantitative data are described as mean ± s.d. for $N \geq 3$, except for some human fetal tissue data which are described as $N = 2$, without s.d. No data were excluded from the analyses and the experiments were not randomized. No statistical method was used to predetermine sample size. The investigators were not blinded to allocation during experiments and outcome assessment. Statistical analysis was performed using a two-tailed unpaired Student's *t*-test using GraphPad Prism 9 software (GraphPad Software). Data distribution was assumed to be normal but this was not formally tested. *P* values lower than 0.05 were considered statistically significant.

### Human iPS cell culture
The feeder-independent induced pluripotent stem (iPS) cell line used for this study was a gift from Silvia Cappello (Max Planck Institute of Psychiatry). Cells were reprogrammed from NuFF3-RQ human newborn foreskin feeder fibroblasts (GSC-3404, GlobalStel)[47]. iPS cells were cultivated as colonies on vitronectin-coated B3 dishes, using mTser medium (STEMCELL Technologies). Colonies were cleaned daily under a binocular stereo microscope (Lynx EVO, Vision Engineering), by manually removing differentiated cells with a needle.

### Cerebral organoids culture
Cerebral organoids were derived from human iPS cells, following a previously published protocol[42]. Day 0 to day 4: iPS colonies of 1–2 mm diameter were detached with pre-warmed collagenase (1 mg ml⁻¹) for 45 min at 37 °C. After the addition of 1 ml of mTser, floating colonies were transferred with a cut tip into a 15 ml tube for two series of gentle washing with medium 1 (DMEM-F12 without phenol red, 20% KOSR, 1× GlutaMAX, 1× MEM-NEAA, 1× 2-Mercaptoethanol, Pen/Strep, 2 µM Dorsomorphin, 2 µM A-83). Colonies were subsequently distributed in an ultra-low attachment 6-well plate with 3 ml of medium 1 and cultivated at 37 °C, 5% CO₂. Day 5–6: half of medium 1 was replaced daily with medium 2 (DMEM-F12 without phenol red, 1× N₂ supplement, 1× GlutaMAX, 1× MEM-NEAA, Pen/Strep, 1 µM CHIR-99021, 1 µM SB-421542). Day 7–14: at day 7, embryoid bodies (EBs) were embedded in Matrigel diluted in medium 2 at a ratio of 2:1. The Matrigel–EB mixture was then spread in an ultra-low attachment dish and incubated at 37 °C for 30 min to solidify (10–20 EBs per well). Finally, medium 2 was gently added to the well, without disturbing the Matrigel patch. On day 14, Matrigel was mechanically broken by pipetting with a 5 ml pipette and transferred into a 15 ml tube for gentle washing. Organoids were suspended in medium 3 (DMEM-F12 without phenol red, 1× N₂ supplement, 1× b27 supplement, 1× GlutaMAX, 1× MEM-NEAA, 1× 2-mercaptoethanol, Pen/Strep, 2.5 µg ml⁻¹ insulin) and grown in ultra-low attachment 6-well plates under agitation at 100 rpm (Digital Orbital Shaker DOS-10M from ELMI). Day 35 to 84: starting from day 35, medium 3 was supplemented with diluted Matrigel (1:100)[48].

### Infection of human fetal cortex and cerebral organoids
Fresh tissue from human fetal cortex was obtained from autopsies performed at the Robert Debré Hospital, and Necker enfants malades Hospital (Paris). Tissues came from spontaneous miscarriages or pregnancy terminations due to kidney malformations. A piece of pre-frontal cortex was collected from one hemisphere, and transported on ice from the hospital to the lab. The tissue was divided into smaller pieces and embedded 4% low-gelling agarose (Sigma) dissolved in artificial cerebrospinal fluid (ACSF). Cerebral organoids (weeks 8–12) were embedded in 3% low-gelling agarose. Gel blocks from both tissues were then sliced with a Leica VT1200S vibratome (300-µm-thick slices) in ice-cold ACSF. Slices were infected with a GFP-coding retrovirus, diluted in DMEM-F12. After 2 h of incubation, slices were washed three times with DMEM-F12 and grown on Millicell cell culture inserts (Merck) in cortical culture medium (DMEM-F12 containing B27, N₂, 10 ng ml⁻¹

fibroblast growth factor (FGF), 10 ng ml⁻¹ epidermal growth factor (EGF), 5% fetal bovine serum and 5% horse serum) for up to 5 days for human fetal brain and 48 h for cerebral organoids. The medium was changed every day. Viruses were produced from HEK-Phoenix-GP cell line, obtained from ATCC (CRL-3215).

### Live imaging in cerebral organoids and human fetal cortex slices
To follow bRG cell divisions for approximately 48 h, we used the following approach. At 48 h after infection (3–5 days for human fetal brain), slices were placed under the microscope by transferring the culture inserts in a 35 mm FluoroDish (WPI) with 1 ml of cortical culture medium (DMEM-F12 containing B27, N₂, 10 ng ml⁻¹ FGF, 10 ng ml⁻¹ EGF, 5% fetal bovine serum and 5% horse serum). Live imaging was performed on a spinning disk wide microscope equipped with a Yokogawa CSU-W1 scanner unit to increase the field of view and improve the resolution deep in the sample. The microscope was equipped with a high working distance (WD 6.9-8.2 mm) ×20 Plan Fluor ELWD NA 0.45 dry objective (Nikon), and a Prime95B SCMOS camera. Z-stacks of 80–100 µm range were taken with an interval of 4–5 µm, and maximum projections were performed. Videos were mounted in Metamorph 7.10. Image treatments (maximum projections, subtract background, median filter, stackreg and rotation) were carried out on Fiji. Data was analysed using Prism 9. Figures were assembled with Affinity Designer 1.9.

### Immunostaining of brain slices
Human fetal brain and cerebral organoid slices in culture were fixed in 4% paraformaldehyde (PFA) for 2 hours. Slices were boiled in sodium citrate buffer (10 mM, pH 6) for 20 minutes and cooled down at room temperature (antigen retrieval). Slices were then blocked in phosphate-buffered saline (PBS)-Triton 100×0.3%-donkey serum 2% at room temperature for 2 hours, incubated with primary antibody overnight at 4 °C in blocking solution, washed in PBS-Tween 0.05%, and incubated with secondary antibody overnight at 4 °C in blocking solution before final wash and mounting in Aquapolymount. Mosaics (tile scans) of fixed tissue were acquired with a CFI Apo LWD Lambda S ×40 objective (WI NA 1.15 WD 0.61–0.59, Nikon).

### Live and fixed correlative microscopy analysis
The correlative microscopy method enables to automatically pair and align live and fixed samples, for cell–cell matching. The macro, based on ImageJ[49] and MATLAB, enables automated contouring of the slices, matching of the live and fixed samples based on their area and shape, and alignment of the samples (rotation and flip if needed). This leads to the precise positioning of the live imaged cells on the immunostained images. This method is described in detail in the Supplementary Information.

### Mathematical model
The model considers three different cell types: bRG cells are type A, IP are type B and neurons are type C. The number of each cell type after *x* division is written $I_x$ (with $I$ = A, B, C) and the probability of producing a cell of type *I* and a cell of type *J* after a bRG division is written $p_{ij}$. The average number of the different cell types after *x* divisions satisfies the recurrence relations:

$$A_x = A_{x-1}(2p_{AA} + p_{AB} + p_{AC}), B_x = B_{x-1} + A_{x-1}(p_{AB} + 2p_{BB}), C_x$$
$$= C_{x-1} + A_{x-1}(p_{AC} + 2p_{CC})$$

Therefore, after *x* divisions, the average number of cells of each cell type is

$$A_x = A_0 \overline{p_A}^x, B_x = A_0 \overline{p_B} \frac{\overline{p_A}^x - 1}{\overline{p_A} - 1}, C_x = A_0 \overline{p_C} \frac{\overline{p_A}^x - 1}{\overline{p_A} - 1},$$

with

$$\overline{p_A} = 2p_{AA} + p_{AB} + p_{AC}, \overline{p_B} = p_{AB} + 2p_{BB}, \overline{p_C} = p_{AC} + 2p_{CC}.$$

The number of bRG cells increases exponentially with the number of division if $\overline{p_A} > 1$ and decreases if $\overline{p_A} < 1$. In the former case, the ratio of non-bRG to bRG cells reaches a constant value $\frac{IP+neurons}{bRG} = \frac{2-\overline{p_A}}{\overline{p_A}-1}$.

## Retrovirus production

To improve transfection efficiency, we used the HEK-Phoenix-GP cell line that stably expresses the packaging enzymes GAL and POL. Cells were plated in 3×T300 (dilution at 1:20) and grown for 3 days to reach 70% of confluence in DMEM-GlutaMax medium, 10% fetal bovine serum (FBS) (50 ml per flask). At day 3, cells were transfected with envelope VSVG plasmid and transfer plasmid (CAG-GFP or MSCV-IRES-GFP) using Lipofectamine 2000. The two plasmids were mixed into 5.4 ml of OptiMEM medium (18 µg E-plasmid/49.5 µg t-plasmid). Lipofectamine 2000 (337.5 µl) was diluted in 5.4 ml of OptiMEM medium and incubated 5 min at room temperature. The DNA preparation was thoroughly mixed into the Lipofectamine preparation and incubated for 30 min at room temperature. In the meantime, medium was changed by 30 ml of DMEM-Glutamax (without FBS) per T300 flask. The DNA–Lipofectamine mixture (3.6 ml) was then added to each T300 flask and incubated for 5 h in a 37 °C incubator. After this period, flasks were carefully transferred into an L3 lab and the medium was changed for 30 ml of fresh DMEM-GlutaMAX, 10% FBS. At day 5, medium was harvested into 50 ml tubes and replaced by 30 ml of fresh medium (samples were stored at 4 °C). At day 6, medium was harvested, pooled with day 5 samples and spun-down to pellet cell debris (1,300 rpm, 5 min at 4 °C). Supernatant was then filtered using 0.22 µm filter unit and divided into six Ultra-Clear tubs (Beckman Coulter, 344058). Tubes were ultra-centrifuged at 31,000g for 90 min at 4 °C. Supernatant was removed, retroviruses were collected with multiple PBS washings and transferred into a single new Ultra-Clear tub. Final ultra-centrifugation was performed (31,000g for 90 min at 4 °C), supernatant was carefully removed and the thin pellet of retroviruses was suspended into 750 µl of DMEM-F12 medium, aliquoted (50 to 100 µl aliquots) and stored at −80 °C. Titre of the preparation was tested by infecting regular HEK cells at different dilution and the proportion of GFP+ cells was measured by FACS (fluorescence-activated cell sorting).

## Notch inhibition experiments

For Notch inhibition experiments, 250-µm-thick organoid slices were infected with a GFP-expressing retrovirus as described above. After infection, slices were transferred to Millicell cell culture inserts (Merck) and placed in a 6-well plate containing cortical culture medium supplemented with 5 µM of (24-diamino-5-phenylthiazole) DAPT (Tocris, 2364). Culture medium was refreshed every day. After 48 hours, organoid slices were fixed, and immunostaining was performed. Cortical culture medium was supplemented with DMSO for the control condition.

## Expression constructs and antibodies

The following plasmids were used in this study: CAG-GFP (a gift from V. Borrell); MSCV-IRES-GFP (Tannishtha Reya, Addgene 20672); VSVG (a gift from P. Benaroch). Antibodies used in this study were mouse anti-SOX2 (Abcam Ab79351, 1/500), sheep anti-EOMES (R&D Sytems AF6166, 1/500), rabbit anti-NEUN (Abcam Ab177487, 1/500), chicken anti-GFP (Abcam Ab13970, 1/500), mouse anti-pVimentin (Abcam Ab22651, 1/1000), rat anti-HES1 (MBL D134-3, 1/500), rabbit anti-NeuroD2 (Abcam, ab104430, 1/500), mouse anti-HuC/HuD (ThermoFisher Scientific, A-21271, 1/200), rabbit anti-HOPX (Proteintech, 11419-1-AP, 1/500), mouse anti-S100β (Synaptic systems 287111, 1/500), mouse anti-OLIG2 (Millipore MABN50, 1/200), mouse anti-LIFR (Abcam 89792, 1/50), rabbit anti-PTPRZ1 (Sigma HPA015103 (Atlas antibodies), 1/500).

Secondary antibodies used were: Donkey Anti-Sheep IgG H&L (Alexa Fluor 405) Abcam ab175676; DyLight 405 AffiniPure Donkey Anti-Mouse IgG (H+L) Jackson ImmunoResearch 715-475-150; DyLight 405 AffiniPure Donkey Anti-Rabbit IgG (H+L) Jackson Immuno-Research 711-475-152; Alexa Fluor 488 AffiniPure Donkey Anti-Rabbit IgG (H+L) Jackson ImmunoResearch 711-545-152; Donkey anti-Rabbit IgG (H+L) Highly Cross-Adsorbed Secondary Antibody, Alexa Fluor Plus 488 Thermo Fisher A32790; Donkey anti-Chicken IgY (H+L) Highly Cross-Adsorbed Secondary Antibody, Alexa Fluor 488 Thermo Fisher A78948; Alexa Fluor 488 AffiniPure Donkey Anti-Chicken IgY (IgG) (H+L) Jackson ImmunoResearch 703-545-155; Alexa Fluor 488 AffiniPure Donkey Anti-Mouse IgG (H+L) Jackson ImmunoResearch 715-545-150; Donkey anti-Mouse IgG (H+L) Highly Cross-Adsorbed Secondary Antibody, Alexa Fluor Plus 488 Thermo Fisher A32766; Donkey anti-Sheep IgG (H+L) Cross-Adsorbed Secondary Antibody, Alexa Fluor 568 Thermo Fisher A21099; Cy3 AffiniPure Donkey Anti-Sheep IgG (H+L) Jackson ImmunoResearch 713-165-147; Donkey anti-Mouse IgG (H+L) Highly Cross-Adsorbed Secondary Antibody, Alexa Fluor 568 Thermo Fisher A10037; Cy3 AffiniPure Donkey Anti-Mouse IgG (H+L) Jackson ImmunoResearch 715-165-150; Donkey anti-Rabbit IgG (H+L) Highly Cross-Adsorbed Secondary Antibody, Alexa Fluor 568 Thermo Fisher A10042; Cy3 AffiniPure Donkey Anti-Rabbit IgG (H+L) Jackson ImmunoResearch 711-165-152; Donkey anti-Rabbit IgG (H+L) Highly Cross-Adsorbed Secondary Antibody, Alexa Fluor Plus 647 Thermo Fisher A32795; Alexa Fluor 647 AffiniPure Donkey Anti-Rabbit IgG (H+L) Jackson ImmunoResearch 715-605-152; Donkey anti-Goat IgG (H+L) Highly Cross-Adsorbed Secondary Antibody, Alexa Fluor Plus 647 Thermo Fisher A32849; Alexa Fluor 647 AffiniPure Donkey Anti-Goat IgG (H+L) Jackson ImmunoResearch 705-605-003; Alexa Fluor 647 AffiniPure Donkey Anti-Mouse IgG (H+L) Jackson ImmunoResearch 715-605-150.

## Ethics statement

Human fetal tissue samples were collected with previous patient consent and in strict observance of legal and institutional ethical regulations. The protocol was approved by the French biomedical agency (Agence de la Biomédecine, approval number: PFS17-003). The procedure is described in detail in the 'Infection of human fetal cortex and cerebral organoids' section of the methods. No participant was compensated in this study.

## Reporting summary

Further information on research design is available in the Nature Portfolio Reporting Summary linked to this article.

## Data availability

The live imaging and immunofluorescence data that support the findings of this study are available on request from the corresponding authors or from C.L. (coquand.laure@gmail.com). Source data are provided with this paper.

## Code availability

The LiveFixedCorrelative code is available at http://xfer.curie.fr/get/lhGgGtXKbHF/Codes_correlative.zip.

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

## Acknowledgements

We acknowledge Institut Curie, a member of the French National Research Infrastructure France-BioImaging (ANR10-INBS-04), and the Nikon BioImaging Center. We thank F. Francis and X. Morin for helpful discussions and critical reading of the paper. L.C. and R. W. were funded

by the French ministry of research (MESRI) from Sorbonne Université. C.B.A. and R. W. were funded by the Fondation pour la Recherche Médicale (FRM). A.-S.M. was funded by the Labex Cell(n)Scale and Centre National de la Recherche Scientifique (CNRS). A.D.B. is an Institut National de la santé et de la recherche médicale (Inserm) researcher. This work was supported by the CNRS, Institut Curie, the ANR (ANR-20-CE16-0004-01) and the Ville de Paris "Emergences" programme.

## Author contributions

L.C. and C.B.A. performed experiments, analysed data and wrote the manuscript. A.-S.M. coded the LiveFixedCorrelative macro, S.F., A.D.C. and M.L. generated organoids, B.B., T.A.-B. and F.G. provided fetal tissue, R.W. analysed data, V.F. assisted with imaging, P.S. generated the mathematical model, J.-B.B. designed the project, performed experiments and analysed data, and A.D.B. designed the project and wrote the manuscript.

## Competing interests

The authors declare no competing interests.

## Additional information

**Extended data** is available for this paper at https://doi.org/10.1038/s41556-024-01393-z.

**Correspondence and requests for materials** should be addressed to Clarisse Brunet Avalos or Alexandre D Baffet.

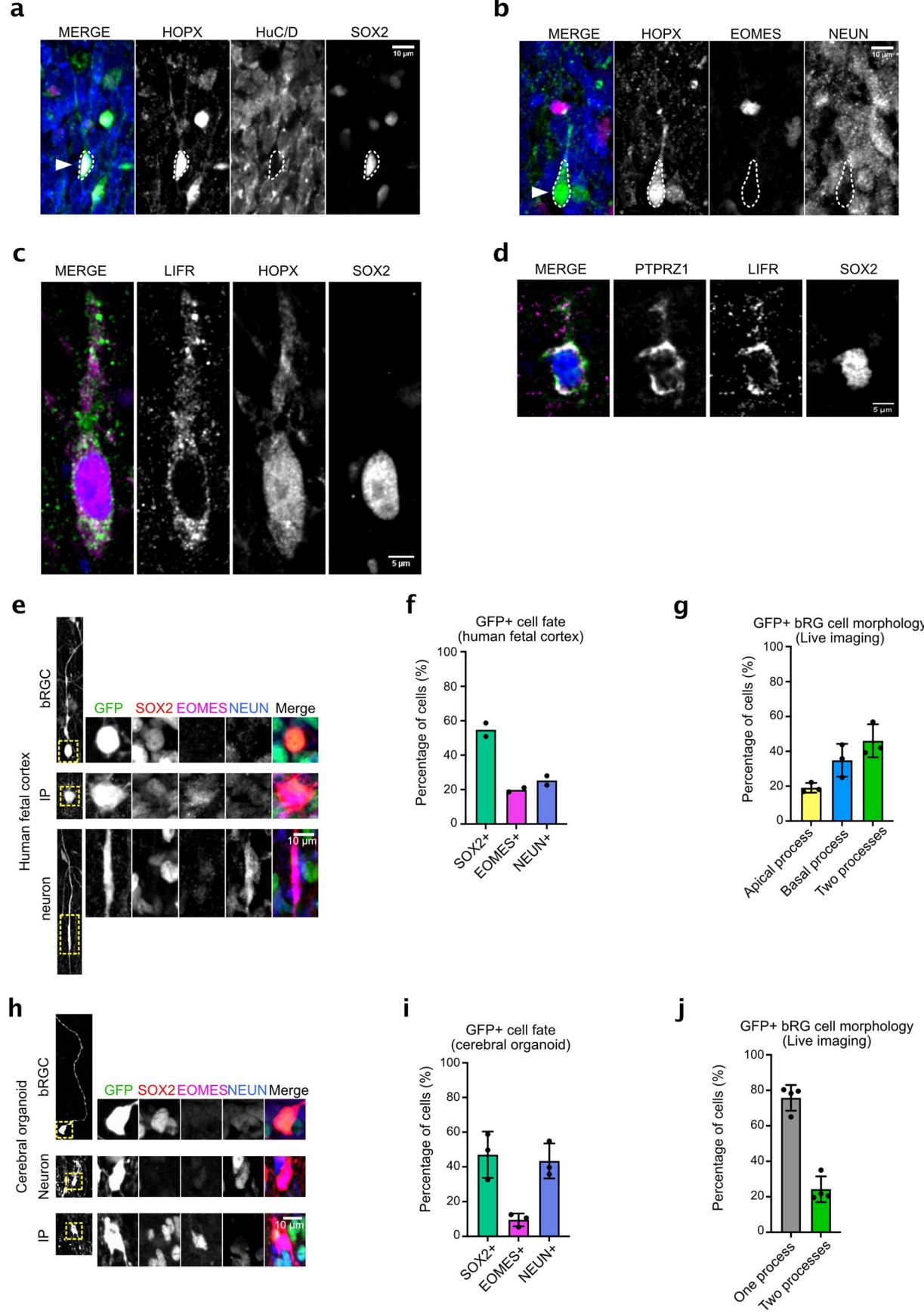

**Extended Data Fig. 1 | See next page for caption.**

**Extended Data Fig. 1 | Fate and shape of cells in cerebral organoids and fetal tissue. a**, Immunostaining for HOPX, HuC/D and SOX2 in human fetal cortex at GW18. **b**, Immunostaining for HOPX, EOMES and NEUN in human fetal cortex at GW18. **c**, Immunostaining for LIFR, HOPX and SOX2 in cortical organoids at week 15. **d**, Immunostaining for PTPRZ1, LIFR and SOX2 in cortical organoids at week 15. **e**, Immunostaining for SOX2, EOMES and NEUN in GFP-infected human fetal cortex at GW17. **f**, Fate of GFP+ cells in human fetal cortex at GW 14-18 (N = 2 brains, 245 cells). **g**, Morphology of GFP+ bRG cells in live imaged human fetal samples at GW 14-18 (N = 3 brains, 284 cells). **h**, Immunostaining for SOX2, EOMES and NEUN in GFP-infected cerebral organoids at week 8. **i**, Fate of GFP+ cells in cerebral organoids at week 8-10 (N = 3 organoid batches, 176 cells). **j**, Morphology of GFP+ bRG cells in live imaged cerebral organoids at week 8-10 (N = 3 organoid batches, 205 cells). Data are presented as mean values +/- SD. All images are representative examples of experiments performed in at least 3 independent organoid batches and 2 independent fetal brains.

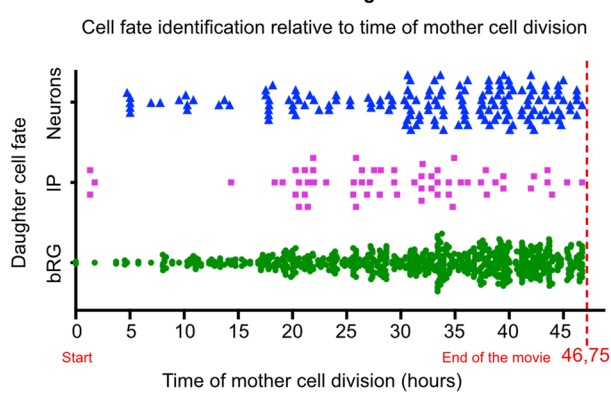

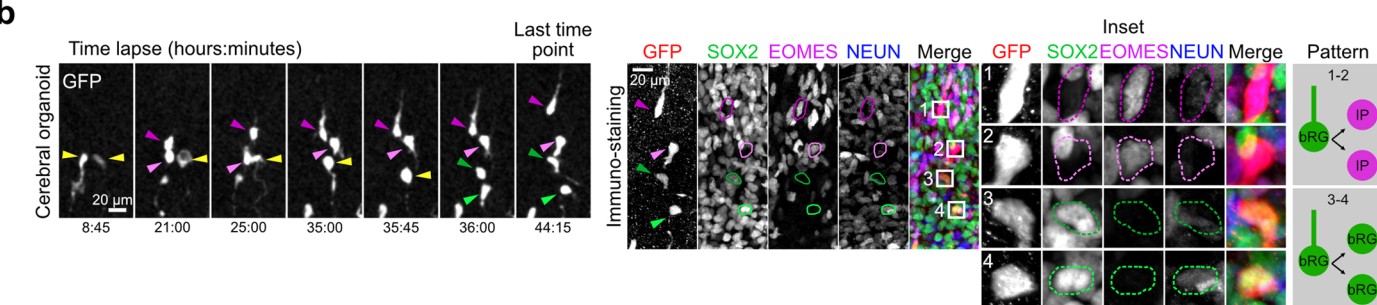

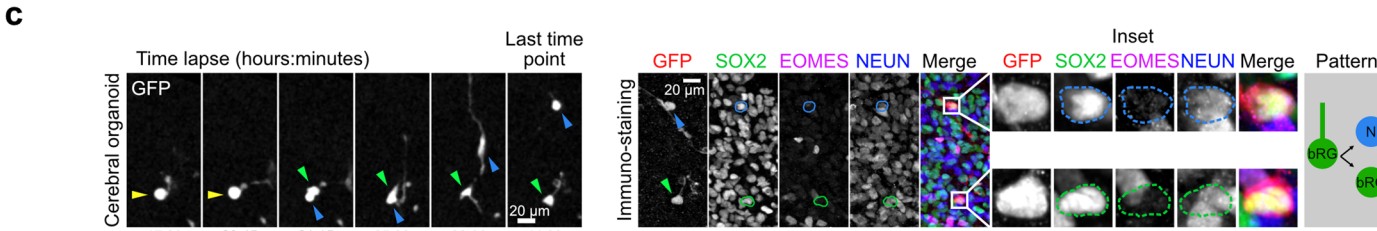

**Extended Data Fig. 2 | Live/fixed correlative examples and cell fate identification timing in cerebral organoids. a**, Detection of bRG, IP or neuronal cell fate relative to the time of division of the bRG mother cell in cerebral organoids at week 8-10 (873 cells from N = 5 organoid batches). **b**, (Top) Live/ fixed correlative analysis of a dividing bRG cell generating two IP daughters.

(Bottom). Live/fixed correlative analysis of a dividing bRG cell generating two bRG daughters. **c**, Live/fixed correlative analysis of a dividing bRG cell generating a bRG daughter and a neuronal daughter. Data are presented as mean values +/- SD. All images are representative examples of experiments performed in at least 3 independent organoid batches.

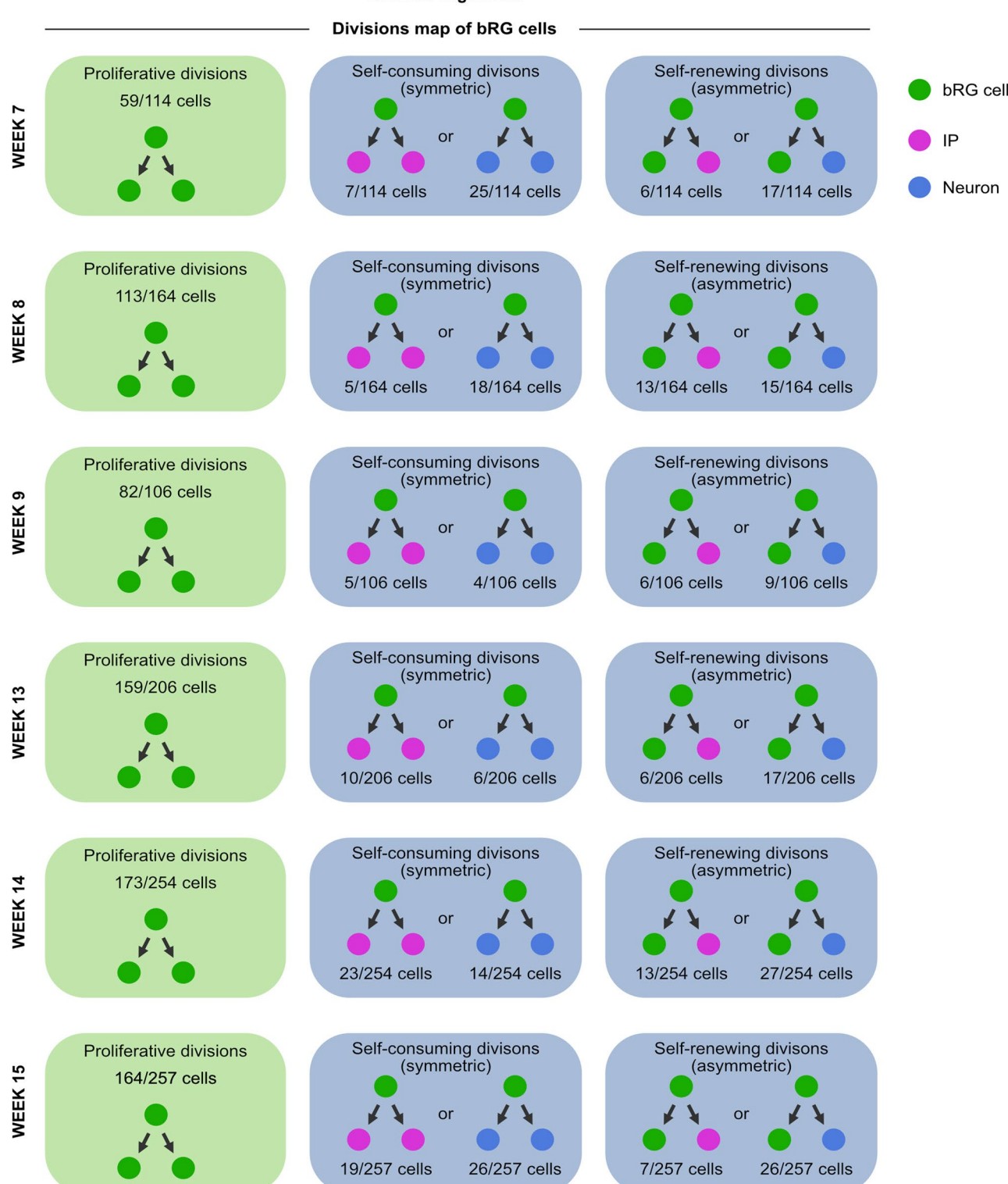

**Extended Data Fig. 3 | Cell fate decision patterns in W7-15 cerebral organoids.** Summary of all division patterns identified in bRG cells in week 7, 8, 9, 13, 14 and 15 cerebral organoids. Week 7 (N = 114 bRG cells), week 8 (N = 164 bRG cells), week 9 (N = 106 bRG cells), week 13 (N = 206 bRG cells), week 14 (N = 254 bRG cells) and week 15 (N = 257 bRG cells).

**a**

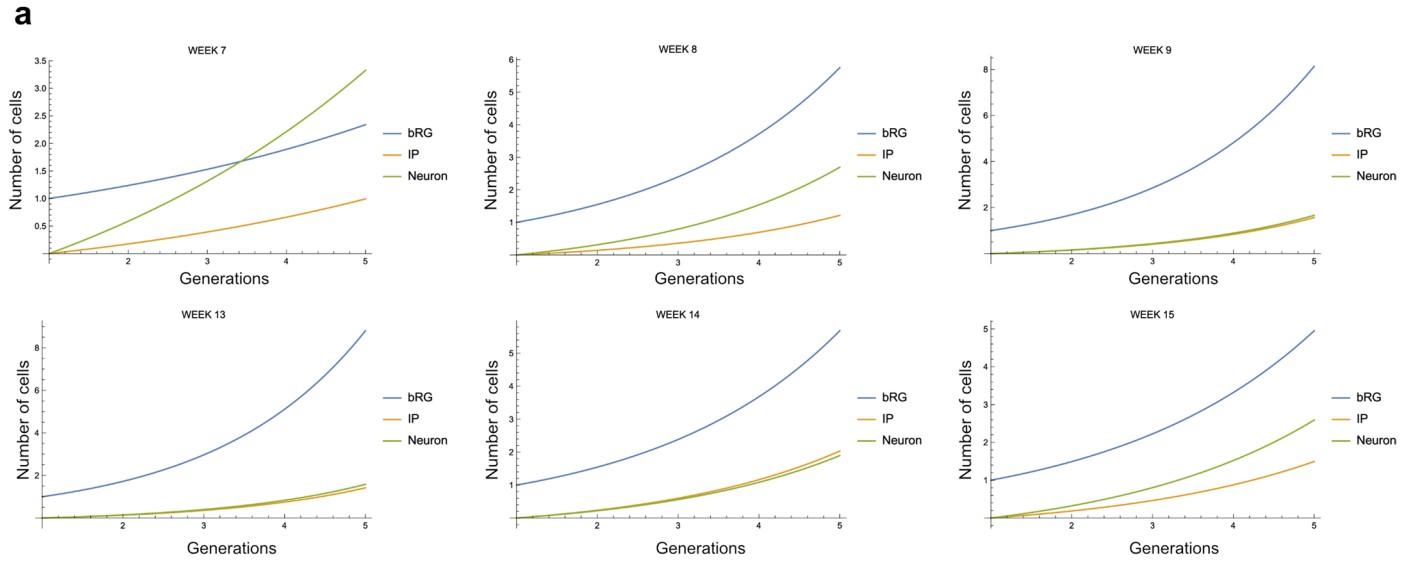

**b**

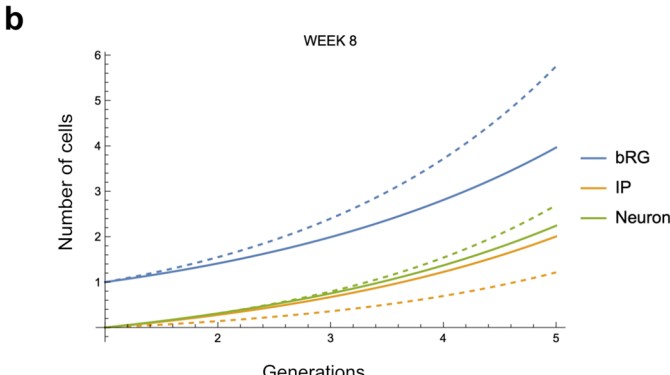

**Extended Data Fig. 4 | bRG cells output in week 7-15 cerebral organoids.**
**a**, Simulation of the output of a single bRG cell after 1-5 generations, in week 7-9 and 13-15 cerebral organoids. **b**, Simulation of the output of a single bRG cell after 1-5 generations in week 8 cerebral organoids (dashed lines) compared to the output of a single bRG cell that underwent 20% less symmetric amplifying divisions in favour of asymmetric indirect divisions (full lines).

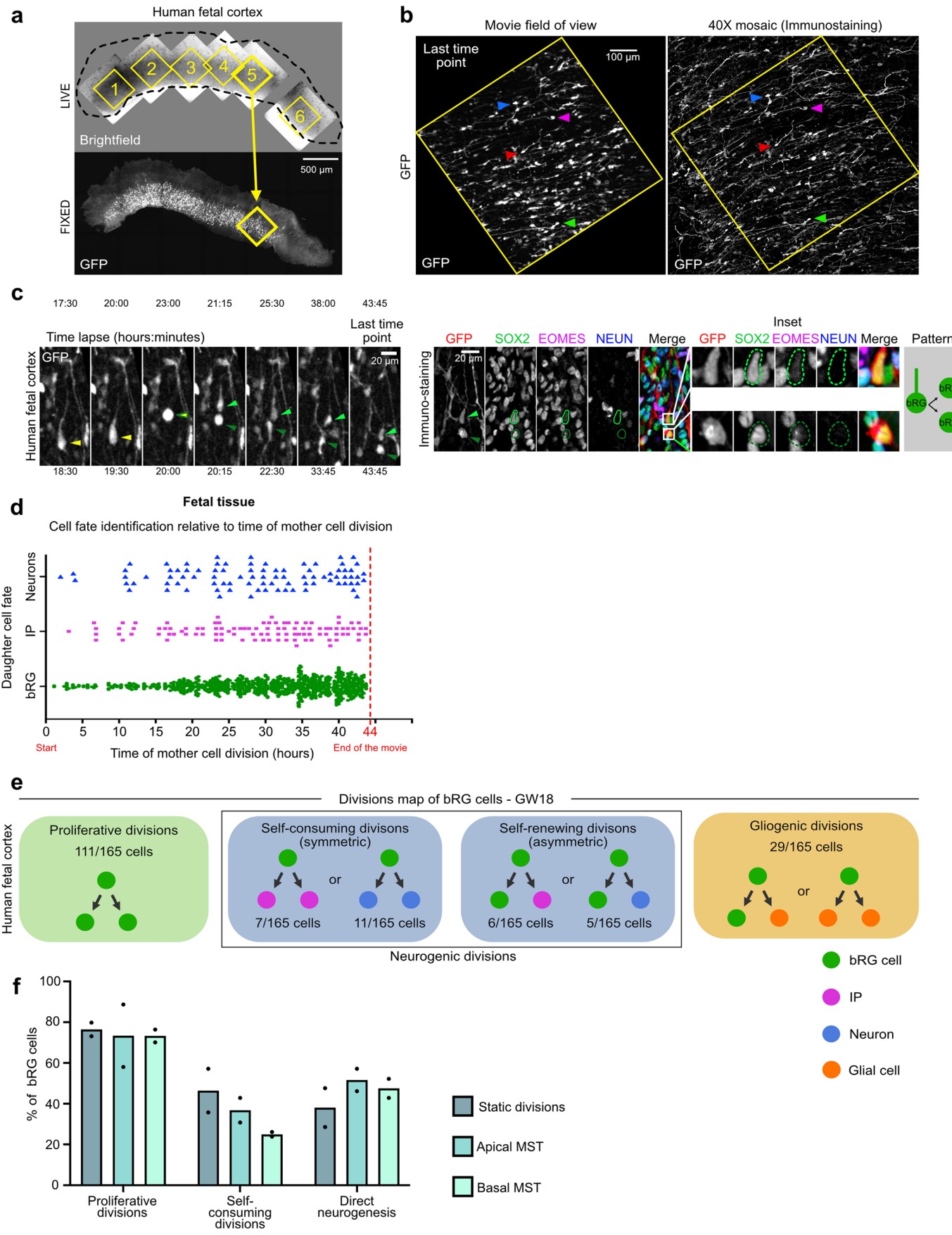

**Extended Data Fig. 5 | See next page for caption.**

**Extended Data Fig. 5 | Cell fate decisions, timing and division modes in fetal tissue. a**, Automated pairing of live and fixed samples and annotation of the video fields of view on the immunostained fixed samples. **b**, GFP+ cell matching between the live images and the fixed images. Arrowheads indicate equivalent cells. **c**, Live/fixed correlative analysis of a dividing bRG cell generating two bRG daughters. **d**, Detection of bRG, IP or neuronal cell fate relative to the time of division of the bRG mother cell in human fetal samples at GW 14-18 (1058 cells from N = 2 fetal brains). **e**, Summary of all division patterns identified in bRG cells at GW 18 (N = 165 bRG cells) human frontal cortex. **f**, Percentage of bRG cells performing proliferative divisions, of neurogenic bRG cells performing self-consuming divisions and of neurogenic bRG cells performing direct neurogenic divisions, depending on their division mode (static, apical MST or basal MST) (N = 2 fetal brains, 415 cells). Data are presented as mean values. All images are representative examples of experiments performed in at least 2 independent fetal brains.

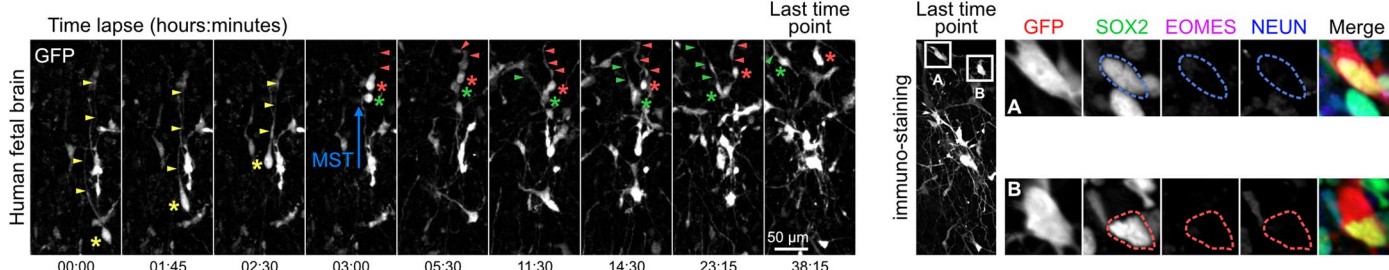

**Extended Data Fig. 6 | SOX2+bRG daughter cells regrow a basal process if at birth.** Live/fixed correlative analysis of a dividing bRG cell generating two bRG daughters. Asterix indicates cell soma and arrowhead indicates basal process. Mother cell (yellow) divides into a process-inheriting cell (red) and a cell that regrows a basal process (green). Representative example of an experiments performed in 3 independent organoid batches.

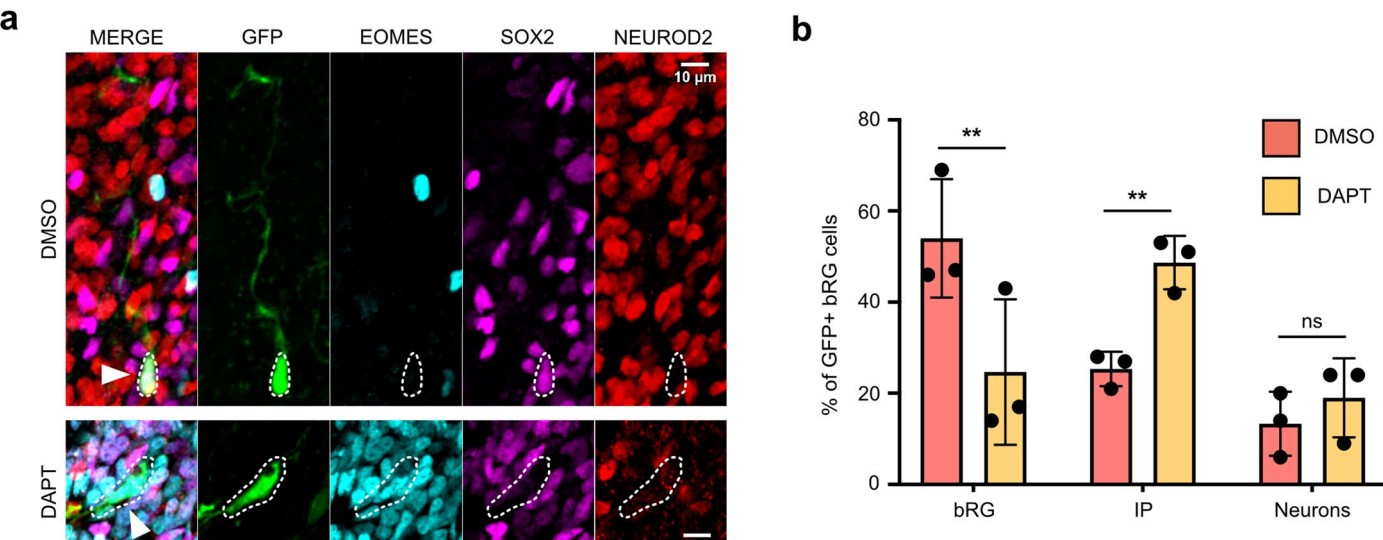

**Extended Data Fig. 7 | Notch inhibition induces RG depletion and IP generation. a**, Immunostaining for SOX2, EOMES and NEUROD2 in GFP-infected week 8 cerebral organoids, following incubation with DMSO or 5 µM DAPT for 48 hours. **b**. Percentage of bRG (SOX2 + ) (p = 0,0047), IPs (EOMES + ) (p = 0,0026) and Neurons (NEUROD2 + ) (p = 0,1221) newborn GFP+ cells after 48 hour treatment with DMSO or 5 µM DAPT (N = 3 organoid batches, 2422 cells). Data are presented as mean values +/- SD. **p < 0,01, by two-tailed t-test.

# Reporting Summary

## Statistics

For all statistical analyses, confirm that the following items are present in the figure legend, table legend, main text, or Methods section.

| n/a | Confirmed | |
|---|---|---|
| ☐ | ☒ | The exact sample size (*n*) for each experimental group/condition, given as a discrete number and unit of measurement |
| ☐ | ☒ | A statement on whether measurements were taken from distinct samples or whether the same sample was measured repeatedly |
| ☐ | ☒ | The statistical test(s) used AND whether they are one- or two-sided<br>*Only common tests should be described solely by name; describe more complex techniques in the Methods section.* |
| ☒ | ☐ | A description of all covariates tested |
| ☒ | ☐ | A description of any assumptions or corrections, such as tests of normality and adjustment for multiple comparisons |
| ☐ | ☒ | A full description of the statistical parameters including central tendency (e.g. means) or other basic estimates (e.g. regression coefficient) AND variation (e.g. standard deviation) or associated estimates of uncertainty (e.g. confidence intervals) |
| ☐ | ☒ | For null hypothesis testing, the test statistic (e.g. *F*, *t*, *r*) with confidence intervals, effect sizes, degrees of freedom and *P* value noted<br>*Give P values as exact values whenever suitable.* |
| ☒ | ☐ | For Bayesian analysis, information on the choice of priors and Markov chain Monte Carlo settings |
| ☒ | ☐ | For hierarchical and complex designs, identification of the appropriate level for tests and full reporting of outcomes |
| ☒ | ☐ | Estimates of effect sizes (e.g. Cohen's *d*, Pearson's *r*), indicating how they were calculated |

*Our web collection on statistics for biologists contains articles on many of the points above.*

## Software and code

Policy information about availability of computer code

| Data collection | Metamorph 7.10 |
|---|---|
| Data analysis | Fiji, Matlab R2023B and Prism 9. The LiveFixedCorrelative code can be downloaded at https://xfer.curie.fr/get/mBUYU6SjQ6T/LiveFixedCorrelative%20Code.zip |

For manuscripts utilizing custom algorithms or software that are central to the research but not yet described in published literature, software must be made available to editors and reviewers. We strongly encourage code deposition in a community repository (e.g. GitHub). See the Nature Portfolio guidelines for submitting code & software for further information.

## Data

Policy information about availability of data

All manuscripts must include a data availability statement. This statement should provide the following information, where applicable:

- Accession codes, unique identifiers, or web links for publicly available datasets
- A description of any restrictions on data availability
- For clinical datasets or third party data, please ensure that the statement adheres to our policy

The live imaging and immunofluorescence data that support the findings of this study are available from the corresponding author (alexandre.baffet@curie.fr) or from the first author (coquand.laure@gmail.com) upon request. The LiveFixedCorrelative code is available at https://xfer.curie.fr/get/mBUYU6SjQ6T/LiveFixedCorrelative%20Code.zip

March 2021

# Human research participants

Policy information about studies involving human research participants and Sex and Gender in Research.

| | |
|---|---|
| Reporting on sex and gender | Sex and Gender have not been taken into account , as this was out of the scope of this study |
| Population characteristics | N/A (anonymzed post-mortem fetal samples) |
| Recruitment | N/A |
| Ethics oversight | French biomedical agency (Agence de la Biomédecine, approval number: PFS17-003) |

Note that full information on the approval of the study protocol must also be provided in the manuscript.

# Field-specific reporting

Please select the one below that is the best fit for your research. If you are not sure, read the appropriate sections before making your selection.

☒ Life sciences ☐ Behavioural & social sciences ☐ Ecological, evolutionary & environmental sciences

For a reference copy of the document with all sections, see nature.com/documents/nr-reporting-summary-flat.pdf

# Life sciences study design

All studies must disclose on these points even when the disclosure is negative.

| | |
|---|---|
| Sample size | Due to the very complex nature of this live imaging-based method, sample size was limited to 3 replica (and sometimes 2 for human fetal tissues, depending on its availability). |
| Data exclusions | No data was excluded from the analysis |
| Replication | All attempts for replication were successful. Experiments were replicated at least 3 times or twice for human fetal tissue samples |
| Randomization | There was no randomization, as we did not compare conditions in this study, but rather quantified different behaviors within control samples. |
| Blinding | no blinding applied as all experiments were performs in the same control samples. |

# Reporting for specific materials, systems and methods

We require information from authors about some types of materials, experimental systems and methods used in many studies. Here, indicate whether each material, system or method listed is relevant to your study. If you are not sure if a list item applies to your research, read the appropriate section before selecting a response.

## Materials & experimental systems

| n/a | Involved in the study |
|---|---|
| ☐ | ☒ Antibodies |
| ☐ | ☒ Eukaryotic cell lines |
| ☒ | ☐ Palaeontology and archaeology |
| ☒ | ☐ Animals and other organisms |
| ☒ | ☐ Clinical data |
| ☒ | ☐ Dual use research of concern |

## Methods

| n/a | Involved in the study |
|---|---|
| ☒ | ☐ ChIP-seq |
| ☒ | ☐ Flow cytometry |
| ☒ | ☐ MRI-based neuroimaging |

## Antibodies

| | |
|---|---|
| Antibodies used | Antibodies used in this study were mouse anti-SOX2 (Abcam Ab79351, clone 9-9-3, 1/500), sheep anti-EOMES (R&D Sytems AF6166, 1/500), rabbit anti-NEUN (Abcam Ab177487, 1/500), chicken anti-GFP (Abcam Ab13970, 1/500), mouse anti-pVimentin (Abcam Ab22651, clone 4A4, 1/1000), rat anti-HES1 (MBL D134-3, clone NM1, 1/500), rabbit anti-NeuroD2 (Abcam, ab104430, 1/500), mouse anti-HuC/HuD (ThermoFisher Scientific, A-21271, clone 16A11, 1/200), rabbit anti-HOPX (Proteintech, 11419-1-AP, 1/500), mouse anti-S100B (Synaptic systems 287111, clone 86D7E4, 1/500), mouse anti-OLIG2 (Millipore MABN50, clone 211F1.1, 1/200), mouse anti-LIFR (Abcam 89792, clone MM0455-9B23, 1/50), rabbit anti-PTPRZ1 (Sigma HPA015103 (Atlas antibodies), 1/500). Secondary antibodies used were: Donkey Anti-Sheep IgG H&L (Alexa Fluor® 405) Abcam ab175676; DyLight™ 405 AffiniPure™ Donkey |

Anti-Mouse IgG (H+L) Jackson ImmunoResearch 715-475-150; DyLight™ 405 AffiniPure™ Donkey Anti-Rabbit IgG (H+L) Jackson ImmunoResearch 711-475-152; Alexa Fluor® 488 AffiniPure™ Donkey Anti-Rabbit IgG (H+L) Jackson ImmunoResearch 711-545-152; Donkey anti-Rabbit IgG (H+L) Highly Cross-Adsorbed Secondary Antibody, Alexa Fluor™ Plus 488 Thermo Fisher A32790; Donkey anti-Chicken IgY (H+L) Highly Cross Adsorbed Secondary Antibody, Alexa Fluor™ 488 Thermo Fisher A78948; Alexa Fluor® 488 AffiniPure™ Donkey Anti-Chicken IgY (IgG) (H+L) Jackson ImmunoResearch 703-545-155; Alexa Fluor® 488 AffiniPure™ Donkey Anti-Mouse IgG (H+L) Jackson ImmunoResearch 715-545-150; Donkey anti-Mouse IgG (H+L) Highly Cross-Adsorbed Secondary Antibody, Alexa Fluor™ Plus 488 Thermo Fisher A32766; Donkey anti-Sheep IgG (H+L) Cross-Adsorbed Secondary Antibody, Alexa Fluor™ 568 Thermo Fisher A21099; Cy™3 AffiniPure™ Donkey Anti-Sheep IgG (H+L) Jackson ImmunoResearch 713-165-147; Donkey anti-Mouse IgG (H+L) Highly Cross-Adsorbed Secondary Antibody, Alexa Fluor™ 568 Thermo Fisher A10037; Cy™3 AffiniPure™ Donkey Anti-Mouse IgG (H+L) Jackson ImmunoResearch 715-165-150; Donkey anti-Rabbit IgG (H+L) Highly Cross-Adsorbed Secondary Antibody, Alexa Fluor™ 568 Thermo Fisher A10042; Cy™3 AffiniPure™ Donkey Anti-Rabbit IgG (H+L) Jackson ImmunoResearch 711-165-152; Donkey anti-Rabbit IgG (H+L) Highly Cross-Adsorbed Secondary Antibody, Alexa Fluor™ Plus 647 Thermo Fisher A32795; Alexa Fluor® 647 AffiniPure™ Donkey Anti-Rabbit IgG (H+L) Jackson ImmunoResearch 715-605-152; Donkey anti-Goat IgG (H+L) Highly Cross-Adsorbed Secondary Antibody, Alexa Fluor™ Plus 647 Thermo Fisher A32849; Alexa Fluor® 647 AffiniPure™ Donkey Anti-Goat IgG (H+L) Jackson ImmunoResearch 705-605-003; Alexa Fluor® 647 AffiniPure™ Donkey Anti-Mouse IgG (H+L) Jackson ImmunoResearch 715-605-150.

Validation

We validated these commonly-used antibodies, based on localization, expression patterns, and co-localization with other overlapping cell fate markers.
-mouse anti-SOX2 (Abcam Ab79351) manufacturer statement: Suitable for: ICC/IF, WB, Flow Cyt (Intra). Reacts with: Mouse, Human
-Sheep anti-EOMES (R&D Sytems AF6166) manufacturer statement: Detects human EOMES in Western blots
-rabbit anti-NEUN (Abcam Ab177487) manufacturer statement: Suitable for: Flow Cyt (Intra), IHC (PFA fixed), mIHC, IHC-P, WB, ICC/IF, IHC-Fr. Reacts with: Mouse, Rat, Sheep, Goat, Cat, Dog, Human, Zebrafish, Common marmoset
-chicken anti-GFP (Abcam Ab13970) manufacturer statement: Suitable for: WB, ICC/IF. Reacts with: Species independent
-mouse anti-pVimentin (Abcam Ab22651). manufacturer statement: Suitable for: ICC/IF, WB, Flow Cyt (Intra). Reacts with: Mouse, Human
-rat anti-HES1 (MBL D134-3) manufacturer statement: Application: ICC, IHC, IP, WB
-rabbit anti-NeuroD2 (Abcam, ab104430). manufacturer statement: Suitable for: IHC-P, IHC-Fr, WB. Reacts with: Mouse, Human
-mouse anti-HuC/HuD (ThermoFisher Scientific, A-21271). manufacturer statement: Immunocytochemistry (ICC/IF). Published species Avian, Cat, Chicken, Chimpanzee, Fish, Guinea pig, Horse, Human, Lizard, Mouse, Non-human primate, Pig, Rabbit, Rat, Reptile, Rhesus monkey, Rodent, Shark, Sheep, Xenopus, Zebrafish
-rabbit anti-HOPX (Proteintech, 11419-1-AP). manufacturer statement: Published Applications: IF; Tested Reactivity Human, Mouse, Rat
-mouse anti-S100B (Synaptic systems 287111). manufacturer statement: Applications: IF.
-mouse anti-OLIG2 (Millipore MABN50) manufacturer statement: immunohistochemistry: suitable
-mouse anti-LIFR (Abcam 89792) manufacturer statement: Reacts with: Human
-rabbit anti-PTPRZ1 (Sigma HPA015103) manufacturer statement: species reactivity human

# Eukaryotic cell lines

Policy information about cell lines and Sex and Gender in Research

Cell line source(s)

- The feeder-independent iPS cell line used for this study was a gift from Silvia Cappello (Max-Plank Institute of Psychiatry - Munich). Cells were reprogrammed from NuFF3-RQ human newborn foreskin feeder fibroblasts (GSC-3404, GlobalStel)
The HEK-Phoenix-GP cell line was obtained from ATCC (CRL-3215)

Authentication

The iPSC line used in this study was genotyped. The HEK-Phoenix-GP cell line was authenticated by ATCC.

Mycoplasma contamination

Mycoplasma testing were performed weekly and always tested negatively.

Commonly misidentified lines
(See ICLAC register)

No commonly misidentified lines were used in this study.

