## [Peer Review File · Nature Cell Biology]

Peer Review Information

Journal: Nature Cell Biology

Manuscript Title: A cell fate decision map reveals abundant direct neurogenesis in the human developing neocortex

Corresponding author name(s): Dr Alexandre Baffet

Editorial Notes:

Reviewer Comments & Decisions:

Decision Letter, initial version:
--

*Please delete the link to your author homepage if you wish to forward this email to co-authors.

Dear Dr Baffet,

Your manuscript, "A cell fate decision map reveals abundant direct neurogenesis in the human developing neocortex", has now been seen by 3 referees, who are experts in cerebral organoids, tissue development, human neocortex, live imaging (referee 1); cerebral organoids, cortex development (referee 2); and brain organoids, human cortex development, radial glial cells, live imaging (referee 3). As you will see from their comments (attached below) they find this work of potential interest, but have raised substantial concerns, which in our view would need to be addressed with considerable revisions before we can consider publication in Nature Cell Biology.

Nature Cell Biology editors discuss the referee reports in detail within the editorial team, including the chief editor, to identify key referee points that should be addressed with priority. To guide the scope of the revisions, I have listed these points below. I should stress that some of the referees' concerns point to a premature dataset and these points would need to be addressed with experiments and data, and reconsideration of the study for this journal and re-engagement of referees would depend on strength of these revisions.

In particular, it would be essential to:

(A) Further substantiate and investigate the claims concerning symmetric-asymmetric divisions, as indicated by all referees.

Referee #1:

"Line 140: "...We noted that when a daughter cell differentiated (e.g. into an EOMES+ IP), it always retained some expression of the mother cell fate marker (SOX2) (Fig. 2c)". A deeper look into the figures indeed reveals that SOX2 is retained in the differentiating cell – but only in asymmetrical divisions. Can the authors confirm that this is indeed the case, describe it better in the text, and provide their own view on why symmetrical neurogenic divisions do not show retaining transient SOX2 expression?"

"Symmetric vs. asymmetric cell division (Line 179). The finding that ~70% of bRG cells divide symmetrically is dramatic for weeks 8-10 organoids (~ day 70). Can the authors test a different (later) time point to show that this may change? In other words, is it a static rate for bRG cells? or does it change with time as occurs for mouse aRG cells?"

Referee #2:

"Discussion: "Our results indicate that a major trajectory for bRG cells consists of symmetric amplifying divisions, followed by self-consuming divisions that generate neurons directly, independently of IPs."

o There is no basis for this statement in the paper. With the current methods, there is no evidence of temporal ordering or any one division mode that is "followed by" another -- all of the analyses are performed at one timepoint".

Referee #3:

"In rodents, radial glia divisions are predominantly symmetric early on, produce neurons directly later, and produce neurons indirectly later still. It would be interesting and important to determine if the results reported here, a shift in division from symmetric to neurogenic could be followed by a predominance of indirect neurogenesis. This could be explored by quantifying division outcomes across the sampled ages in fetal tissue (GW 14-18), and/or in organoids which could be cultured for longer timepoints".

(B) Address methodological concerns to support major claims, as indicated by referees #1 and #2:

Referee #1:

"bRG cell identity. While the criteria used here to identify bRG cells are thorough, it would be good to get a more direct identity measure such as specific bRG markers. While it is understood that immunostaining for bRG markers such as LIFR1, HOPX and PTPRZ1 may not be conclusive due to the abundance of such markers also in non bRG cells, it would be imperative to show a few cases where co-expression of such markers together in candidate bRG cells is indicated to strengthen the method of bRG cell identification. scRNA-Seq is always the preferred method due the uncertainty of these markers when expressed alone, but again, co-expression at the immunostaining level can do the work. This will also help strengthen the criteria for bRG cell identification in line 140: "double-negative (EOMES-/NEUN-) cells being identified as bRG cells".

Referee #2:

"GW14-17 is a broad range of development, particularly with regards to the emergence and abundance of bRG. Samples spanning these ages should not be directly pooled, or data from individual samples should be shown to allow readers to see sample variation across timepoints".

"The manuscript makes it clear that there are minority populations of these dividing cells that do not conform to generally assumed rules, such as bRG that do not have processes, bRG that undergo stationary divisions, and IPs that have processes. This makes interpretability of some of the later findings challenging when it comes to assigning fates to a progenitor that underwent a consuming division. For example, Figure 2d shows a stationary division that results in generation of two neurons, while Figure 3b shows an MST division that results in generation of two neurons. The mother cell in 3b is labeled as a bRG which seems correct based on the MST division, but it is impossible to assign the mother cell identity in Figure 2d based on the authors' own observation of bRG that undergo stationary divisions. Similarly, Figure 3a shows a stationary division that yields two IPs; the mother cell is assigned as a bRG, but previous works have suggested that IPs may be able to symmetrically divide to make more IPs (Hansen 2010). This assignment of the mother cell has significant implications for the eventual summary figures when it comes to assigning the division modes and outputs of these cell types. Given the shared behavior of stationary division between bRG and IPs and

even the shared existence of processes, it seems that any stationary division that does not result in at least one bRG cannot be confidently assigned to a bRG or IP. The authors state that they analyze bRG divisions that undergo MST, which is reassuring, but this caveat introduces a ~20% error in their summary figures by leaving them unable to count stationary bRG divisions".

(C) Ensure you are using an adequate number of independent experiments and independent biological samples to support conclusions, as noted by:

Referee #2:

"The current study has far too low n of 16 cells to make any conclusion about the necessity of Notch for progenitor fate maintenance. A direct perturbation would increase confidence in the authors' finding and would provide better support for their speculation that Notch signaling and its effect on radial glia is related to the cell soma's microenvironment".

Referee #3:

"My major concern is about Figure 5 which led to an important conclusion of the manuscript but was only done in one sample".

(D) All other referee concerns pertaining to strengthening existing data, providing controls, clarifications and textual changes, should also be addressed.

(E) Finally please pay close attention to our guidelines on statistical and methodological reporting (listed below) as failure to do so may delay the reconsideration of the revised manuscript. In particular please provide:

- a Supplementary Figure including unprocessed images of all gels/blots in the form of a multi-page pdf file. Please ensure that blots/gels are labeled and the sections presented in the figures are clearly indicated.
- a Supplementary Table including all numerical source data in Excel format, with data for different figures provided as different sheets within a single Excel file. The file should include source data giving rise to graphical representations and statistical descriptions in the paper and for all instances where the figures present representative experiments of multiple independent repeats, the source data of all repeats should be provided.

We would be happy to consider a revised manuscript that would satisfactorily address these points, unless a similar paper is published elsewhere, or is accepted for publication in Nature Cell Biology in the meantime.

- ensure that it conforms to our format instructions and publication policies (see below and www.nature.com/nature/authors/).
- provide a point-by-point rebuttal to the full referee reports verbatim, as provided at the end of this

letter.

- provide the completed Editorial Policy Checklist (found here <https://www.nature.com/authors/policies/Policy.pdf>), and Reporting Summary (found here <https://www.nature.com/authors/policies/ReportingSummary.pdf>). This is essential for reconsideration of the manuscript and these documents will be available to editors and referees in the event of peer review. For more information see <http://www.nature.com/authors/policies/availability.html> or contact me.

Nature Cell Biology is committed to improving transparency in authorship. As part of our efforts in this direction, we are now requesting that all authors identified as 'corresponding author' on published papers create and link their Open Researcher and Contributor Identifier (ORCID) with their account on the Manuscript Tracking System (MTS), prior to acceptance. ORCID helps the scientific community achieve unambiguous attribution of all scholarly contributions. You can create and link your ORCID from the home page of the MTS by clicking on 'Modify my Springer Nature account'. For more information please visit www.springernature.com/orcid.

[Redacted]

We would like to receive a revised submission within six months. We would be happy to consider a revision even after this timeframe, however if the resubmission deadline is missed and the paper is eventually published, the submission date will be the date when the revised manuscript was received.

We hope that you will find our referees' comments, and editorial guidance helpful. Please do not hesitate to contact me if there is anything you would like to discuss.

Best wishes,

Stelios

Stylianos Lefkopoulos, PhD
He/him/his
Associate Editor
Nature Cell Biology
Springer Nature
Heidelberger Platz 3, 14197 Berlin, Germany

E-mail: stylianos.lefkopoulos@springernature.com
Twitter: @s_lefkopoulos

Reviewers' Comments:

Reviewer #1:

Remarks to the Author:

Coquand et al. describe a developed a method to quantitatively map human bRG cell division modes in the human developing cortex and in cerebral organoids. The method is based on a semi-automated imaging of live-fixed tissue, which enables identification of bRG cell progeny fate following their division. The authors first identify morphology of bRG cells in human embryonic cortical samples and cerebral organoids based on their location basally to VZ areas, expression of SOX2 and pVim, the presence of at least one process and the typical mitotic soma translocation (MST) following cell division. The authors then devise a semiautomated system to follow dividing cells in live tissue followed by fixation, which is based on computerized screening of multiple videos of live cells and matching those to their corresponding fixated and immunostained counterparts (the same cells). This allows the fate analysis of two daughter cells derived from a bRG cell following division.

Based on these analyses, the authors find high rate of proliferative bRG cells (60%, giving rise to 2 bRG cells) in cerebral organoids – as sign for bRG self-renewal in the oSVZ, while a significant portion of bRG cells was consumed for generating neurons by symmetric divisions producing two neurons. This is in addition to the asymmetric division mode producing one bRG cell and one neuron (both modes either directly or via IP cells). To gain more insights on transition through these modes the authors first detect preferential symmetric neurogenic divisions at more distal / basal regions of the oSVZ, while asymmetric self-renewal or symmetrical/asymmetrical neurogenic divisions appeared comparable along the oSVZ. On the other hand, among those asymmetric neurogenic cell divisions (producing bRG and IP cell/neuron) that can be detected more apically in the oSVZ, no correlation between a bRG daughter cell fate and apical or basal process inheritance was found. This was confirmed in both fetal tissue and organoids. To support this, the authors show no preferential Notch signalling (stem cell state) in bRG cells (before or after cell division) that inherited either an apical or a basal process. The authors conclude that a major trajectory for bRG cells consists of proliferative, symmetric self-renewal of bRG cells followed by self-consuming, symmetric divisions that generate neurons directly, independently of IPs. Finally, they conclude that a stem cell fate in daughter cells is irrespective of basal process inheritance. The authors present these findings in light of the sharp contrast to mouse aRG cells that largely rely on IPs to amplify the neurogenic output and on process inheritance for determining stem cell state.

The main and most outstanding finding in this work, which also provides an important novel advancement towards understanding stem cell maintenance and differentiation trajectories in the developing human cortex is that bRG cells expand the human cortex mainly in symmetrical manner. They do so in two major modes that well balance each other in terms of cell pool quantities: the first is the self-expanding proliferative mode where bRG cells symmetrically divide to produce 2 bRG cells; this is followed by the second, self-consuming neurogenic mode, where bRG cells again divide symmetrically, this time to directly produce two neurons. While these modes of cortical expansion are further accompanied by the more conventional trajectories of asymmetrical divisions generating bRG cells and neurons/IP cells or symmetrical divisions producing two IP cells, this study finds evidence that former expansive/consumptive symmetrical mode is dominating the growth of the human cortex during development when compared to the mouse cortex. This is further added to the fact that radial glial processes do not seem to play any role dictating the stem cell stage within the pool of the asymmetrically dividing bRG cells, providing further room for exploring how human bRG cells exert the oSVZ environment to expand the neurogenic pool in an asymmetric manner. This method is highly not trivial and very well appreciated. The technical work is thorough and well planned and done. The choice of GW 14-17 in vivo and week 8-10 cerebral organoids is well matched. And the finding that asymmetrically dividing bRG cells generating one IP and one neuron was never observed is super interesting. Finally, because the data already provide a shift in the way we understand stem cell proliferation and differentiation trajectories in humans, any “natural requests” to take advantage of this method for comparing the findings on bRG cells to aRG cells, to explore the subtypes of neurons (or even glia) that are produced by these types of bRG modes - all can be simply considered as

beyond the scope of this work. Therefore, this work is highly recommended for publication.

There are several points to be considered in any revised manuscript format:

1. bRG cell identity. While the criteria used here to identify bRG cells are thorough, it would be good to get a more direct identity measure such as specific bRG markers. While it is understood that immunostaining for bRG markers such as LIFR1, HOPX and PTPRZ1 may not be conclusive due to the abundance of such markers also in non bRG cells, it would be imperative to show a few cases where co-expression of such markers together in candidate bRG cells is indicated to strengthen the method of bRG cell identification. scRNA-Seq is always the preferred method due the uncertainty of these markers when expressed alone, but again, co-expression at the immunostaining level can do the work. This will also help strengthen the criteria for bRG cell identification in line 140: "double-negative (EOMES-/NEUN-) cells being identified as bRG cells".

2. Line 140: "...We noted that when a daughter cell differentiated (e.g. into an EOMES+ IP), it always retained some expression of the mother cell fate marker (SOX2) (Fig. 2c)". A deeper look into the figures indeed reveals that SOX2 is retained in the differentiating cell – but only in asymmetrical divisions. Can the authors confirm that this is indeed the case, describe it better in the text, and provide their own view on why symmetrical neurogenic divisions do not show retaining transient SOX2 expression?

3. Symmetric vs. asymmetric cell division (Line 179). The finding that ~70% of bRG cells divide symmetrically is dramatic for weeks 8-10 organoids (~ day 70). Can the authors test a different (later) time point to show that this may change? In other words, is it a static rate for bRG cells? or does it change with time as occurs for mouse aRG cells?

4. Comparison with human fetal tissue: it is not always clear why the authors alternate between fetal and organoid tissue. At some parts, mainly the establishment of the method, it was done in both tissues and was also essential. Other than this, the rationale should be better articulated.

5. A model for division patterns (Fig. 3d). Can the authors calculate a model of cortical expansion from these interesting findings? How many cells are generated that can accommodate for cortical development via these modes compared to before these modes were identified as major in this study (i.e. compare to the use of IP cell rather than direct, and less proliferative symmetrical divisions). In other words, how these modes are advantageous for cortical expansion.

Reviewer #2:
Remarks to the Author:
Summary

Coquand et al. develop a novel pipeline for systematically integrating live imaging data with fixed immunohistochemistry and implement it to track the behaviors and lineages of basal progenitors in the developing cortex. This method represents a very useful tool that can increase the throughput and accuracy of studies of a similar nature, and the authors use it to address an important question in the field of cortical development. The authors show in primary tissue and organoids that basal radial glia undergo both proliferative and neurogenic divisions and note a gradient of direct vs. indirect neurogenesis based on proximity to the apical surface of the developing cortex. While technically impressive, the major findings are not altogether novel and are not very deeply explored. The nature of bRG divisions and the relative frequency of their outputs has been reported in previous studies in human and primate, some of which have gone more in depth by reporting other

factors such as mitotic cleavage angles or by tracing progenitors over multiple divisions to more accurately report their proliferative potential. The association with Notch signaling is quite weak in this study, relying on a single IHC panel and the absence of a positive signal to make a claim, whereas previous studies have more thoroughly and directly interrogated the effect of Notch on cell behaviors and daughter fate. The finding of direct vs. indirect neurogenesis correlating with distance from the apical surface does seem novel and is of significant interest to the field, but there is no further investigation of this phenomenon or mechanistic hypotheses.

Overall, this study presents a highly useful method that could benefit the field and show that its use corroborates an established base of knowledge about basal radial glia dynamics. However, the paper currently falls short of significantly expanding knowledge of these cellular behaviors and offers little mechanistic insight. I would invite the authors to more critically analyze the data that they have and follow up their findings with more direct perturbations in order to set their work above the current standard of the field.

General comments

- GW14-17 is a broad range of development, particularly with regards to the emergence and abundance of bRG. Samples spanning these ages should not be directly pooled, or data from individual samples should be shown to allow readers to see sample variation across timepoints.
- The manuscript makes it clear that there are minority populations of these dividing cells that do not conform to generally assumed rules, such as bRG that do not have processes, bRG that undergo stationary divisions, and IPs that have processes. This makes interpretability of some of the later findings challenging when it comes to assigning fates to a progenitor that underwent a consuming division. For example, Figure 2d shows a stationary division that results in generation of two neurons, while Figure 3b shows an MST division that results in generation of two neurons. The mother cell in 3b is labeled as a bRG which seems correct based on the MST division, but it is impossible to assign the mother cell identity in Figure 2d based on the authors' own observation of bRG that undergo stationary divisions. Similarly, Figure 3a shows a stationary division that yields two IPs; the mother cell is assigned as a bRG, but previous works have suggested that IPs may be able to symmetrically divide to make more IPs (Hansen 2010). This assignment of the mother cell has significant implications for the eventual summary figures when it comes to assigning the division modes and outputs of these cell types. Given the shared behavior of stationary division between bRG and IPs and even the shared existence of processes, it seems that any stationary division that does not result in at least one bRG cannot be confidently assigned to a bRG or IP. The authors state that they analyze bRG divisions that undergo MST, which is reassuring, but this caveat introduces a ~20% error in their summary figures by leaving them unable to count stationary bRG divisions.
- The authors surprisingly seem to ignore one of their major findings of difference between organoids and primary cortex, which is that bRG in the cortical tissue undergo substantially more indirect neurogenesis compared to organoids. Organoid bRG directly generated neurons in 50/70 (71%) self-consuming divisions and in 52/81 (64%) self-renewing divisions, while primary bRG directly generated neurons in only 18/46 (40%) self-consuming divisions and 34/92 (37%) self-renewing divisions. This is quite a striking difference that goes completely underreported and raises serious questions about the claim that primary bRG "closely match" organoid bRG, in addition to obscuring a legitimate question of why organoid bRG are biased towards direct neurogenesis.
- The figures in general are somewhat padded with non-essential information. Figures 2c-e, 3a-c, and 4a-d do not inherently add much more information and are essentially extensions of the methods. Even the stacked barcharts on the right side of Figure 5 offer fairly little new information or synthesis relative to the swarmplots. The figures could be made more concise or introduce more analyses/comparisons rather than repeating n of 1 examples that could be presented in extended data.
- The authors seem to miss an opportunity with this study by limiting their live imaging window to 48 hours. While I understand the technical challenges associated with preserving the samples during the

imaging and generating more data to analyze, it feels like a valuable opportunity to further explore the fate potential of individual cells. Tracking multiple divisions could also provide more insight on some of the observations made from the poster – if the same bRG divides close to the apical surface during one division and then farther away for a second division, is its fate potential affected as would be predicted by the current results? These longer term experiments have been conducted in the past (Betizeau 2013) and it would be of significant interest to the field to validate those findings from macaque in a human context.

- The Notch experiments are extremely limited to a single HES1 staining panel and miss an opportunity to show with direct intervention that Notch activation is relevant to daughter cell fate or division mode. Again, previous work has explored this question by treating with DAPT to inhibit Notch, which resulted in a bias towards IP fate at the expense of bRG fate (Hansen 2010). The current study has far too low n of 16 cells to make any conclusion about the necessity of Notch for progenitor fate maintenance. A direct perturbation would increase confidence in the authors' finding and would provide better support for their speculation that Notch signaling and its effect on radial glia is related to the cell soma's microenvironment.

Specific feedback

- Figure 1b
 - o insets for Zone 1 and Zone 3 do not seem to match the low-mag image
- Figure 1h
 - o It's unclear if Sox2 and EOMES are considered to be mutually exclusive. Past work has shown that these markers can be co-expressed (Hansen 2010), as do images from later in the paper (Figure 4d)
- Figure 5
 - o The stacked bar charts do not offer new insight beyond the swarmplots, and there is no reference to them in the text or explanation of why the seemingly-arbitrary 1mm distance from the apical surface was chosen as a relevant metric to bin cells.
- Figure 6f
 - o The assignment of the bottom cell as a bRG and not a neuron seems quite subjective based on the NeuN staining perfectly colocalizing with the labeled cell, especially when compared to figure 3d which also shows a cell with bright SOX2 and dim NeuN but is called a neuron.
- Figure 7g
 - o The model is very unclear and does not help understand what the authors' main point is. They seem to suggest that there's Notch everywhere in the OSVZ and would be on in IPs as well, but they show in their data that this is not the case.
 - Discussion: "Our results indicate that a major trajectory for bRG cells consists of symmetric amplifying divisions, followed by self-consuming divisions that generate neurons directly, independently of IPs."
 - o There is no basis for this statement in the paper. With the current methods, there is no evidence of temporal ordering or any one division mode that is "followed by" another -- all of the analyses are performed at one timepoint.
 - Discussion: "aRG cells rely on IPs to amplify their neurogenic output, as their own amplification is limited by spatial constraints... bRG cells on the other hand are not subject to this physical limitation and can amplify their own pool both radially and tangentially, and thus IPs are less relied upon to increase the neurogenic output."
 - o This statement is speculative and should be introduced as such. There is little conceptual merit to this idea either – why does producing an IP somehow take up less space than a neuron?

Core findings:

- Figure 1

- o oRG can have different morphologies
- o EOMES+ cells can have processes or not
- o MST direction follows process
- o Organoid RG have similar morphologies and division directions as primary
 - Figure 2
- o Live imaging methodology
- o Able to observe bRG asymmetric division and IPC symmetric division
 - Figure 3
- o Able to observe more division modes
- o Majority of organoid bRG undergo proliferative divisions
- o 50/50 for consuming or asymmetric neurogenic divisions
 - ♣ Consuming divisions are biased towards direct neurogenesis (50/70 generate neurons, not IPCs)
 - ♣ Asymmetric are biased towards direct neurogenesis (52/81 generate one bRG and one neuron, not one IPC)
 - Figure 4
- o Same as figure 3, but for primary instead of organoid
- o Primary is biased towards IPC generation over neuron generation, especially compared to organoids?
 - ♣ This point seems not followed up on, at least in the figures
 - Figure 5
- o No spatial separation for proliferative vs. neurogenic divisions
- o No spatial separation for self-consuming or self-renewing neurogenic divisions
- o Spatial bias for direct vs. indirect neurogenic divisions – direct occur farther away from apical surface, indirect occur closer to apical surface
 - Figure 6
- o Fiber inheritance does not affect cell fate in either organoids or primary
 - Figure 7
- o HES1 is on in a subset of bRG
- o HES1 is never expressed in the differentiating daughter
- o Process inheritance is irrelevant to HES1 expression in asymmetric divisions

Reviewer #3:

Remarks to the Author:

Coquand et al. characterize the proliferation mode of basal radial glia (bRG) in both midgestational primary human cortical slices and cerebral brain organoids. They demonstrate that approximately half of the neurogenic divisions by bRG produce neurons directly without going through an intermediate progenitor cell. Moreover, bRGs at greater distance from the apical surface are more likely to go through direct neurogenic divisions. They also found that inheritance of the basal process does not maintain the fate of daughter cells as bRG.

The manuscript is well prepared, and experiments are well designed/executed. Basal radial glia are an important type of neural stem cell and further characterizing cell behavior and fate will generally contribute to our understanding of cortical expansion in mammals.

I have three comments:

1. My major concern is about Figure 5 which led to an important conclusion of the manuscript but was only done in one sample.
2. If direct neurogenesis is more common in the upper region of the OSVZ, one would predict that

EOMES+ intermediate progenitor cells would be less abundant there relative to radial glia. Could the authors quantify the proportion of radial glia, intermediate progenitor cells, and neurons with respect to their distance from the apical surface in fetal human slices?

3. In rodents, radial glia divisions are predominantly symmetric early on, produce neurons directly later, and produce neurons indirectly later still. It would be interesting and important to determine if the results reported here, a shift in division from symmetric to neurogenic could be followed by a predominance of indirect neurogenesis. This could be explored by quantifying division outcomes across the sampled ages in fetal tissue (GW 14-18), and/or in organoids which could be cultured for longer timepoints.

Minor comment:

We are told the human fetal tissue came from autopsies. The authors should clarify whether these were normotypic or pathologic samples.

Methods should be written concisely, but should contain all elements necessary to allow interpretation and replication of the results. As a guideline, Methods sections typically do not exceed 3,000 words. The Methods should be divided into subsections listing reagents and techniques. When citing previous methods, accurate references should be provided and any alterations should be noted. Information must be provided about: antibody dilutions, company names, catalogue numbers and clone numbers for monoclonal antibodies; sequences of RNAi and cDNA probes/primers or company names and catalogue numbers if reagents are commercial; cell line names, sources and information on cell line identity and authentication. Animal studies and experiments involving human subjects must be reported in detail, identifying the committees approving the protocols. For studies involving human subjects/samples, a statement must be included confirming that informed consent was obtained. Statistical analyses and information on the reproducibility of experimental results should be provided in a section titled "Statistics and Reproducibility".

All Nature Cell Biology manuscripts submitted on or after March 21 2016 must include a Data availability statement at the end of the Methods section. For Springer Nature policies on data availability see <http://www.nature.com/authors/policies/availability.html>; for more information on this particular policy see <http://www.nature.com/authors/policies/data/data-availability-statements-data-citations.pdf>. The Data availability statement should include:

- Accession codes for primary datasets (generated during the study under consideration and designated as "primary accessions") and secondary datasets (published datasets reanalysed during the study under consideration, designated as "referenced accessions"). For primary accessions data

should be made public to coincide with publication of the manuscript. A list of data types for which submission to community-endorsed public repositories is mandated (including sequence, structure, microarray, deep sequencing data) can be found here <http://www.nature.com/authors/policies/availability.html#data>.

- Unique identifiers (accession codes, DOIs or other unique persistent identifier) and hyperlinks for datasets deposited in an approved repository, but for which data deposition is not mandated (see here for details <http://www.nature.com/sdata/data-policies/repositories>).
- At a minimum, please include a statement confirming that all relevant data are available from the authors, and/or are included with the manuscript (e.g. as source data or supplementary information), listing which data are included (e.g. by figure panels and data types) and mentioning any restrictions on availability.
- If a dataset has a Digital Object Identifier (DOI) as its unique identifier, we strongly encourage including this in the Reference list and citing the dataset in the Methods.

We recommend that you upload the step-by-step protocols used in this manuscript to the Protocol Exchange. More details can be found at www.nature.com/protocolexchange/about.

All imaging data should be accompanied by scale bars, which should be defined in the legend. Cropped images of gels/blots are acceptable, but need to be accompanied by size markers, and to retain visible background signal within the linear range (i.e. should not be saturated). The boundaries of panels with low background have to be demarked with black lines. Splicing of panels should only be considered if unavoidable, and must be clearly marked on the figure, and noted in the legend with a statement on whether the samples were obtained and processed simultaneously. Quantitative comparisons between samples on different gels/blots are discouraged; if this is unavoidable, it should only be performed for samples derived from the same experiment with gels/blots were processed in parallel, which needs to be stated in the legend.

- For line art, graphs, charts and schematics we prefer Adobe Illustrator (.AI), Encapsulated PostScript (.EPS) or Portable Document Format (.PDF). Files should be saved or exported as such directly from the application in which they were made, to allow us to restyle them according to our journal house style.
- We accept PowerPoint (.PPT) files if they are fully editable. However, please refrain from adding PowerPoint graphical effects to objects, as this results in them outputting poor quality raster art. Text used for PowerPoint figures should be Helvetica (preferred) or Arial.
- We do not recommend using Adobe Photoshop for designing figures, but we can accept Photoshop generated (.PSD or .TIFF) files only if each element included in the figure (text, labels, pictures, graphs, arrows and scale bars) are on separate layers. All text should be editable in 'type layers' and line-art such as graphs and other simple schematics should be preserved and embedded within 'vector smart objects' - not flattened raster/bitmap graphics.
- Some programs can generate Postscript by 'printing to file' (found in the Print dialogue). If using an application not listed above, save the file in PostScript format or email our Art Editor, Allen Beattie for advice (a.beattie@nature.com).

The total number of Supplementary Figures (not including the “unprocessed scans” Supplementary Figure) should not exceed the number of main display items (figures and/or tables (see our Guide to Authors and March 2012 editorial <http://www.nature.com/ncb/authors/submit/index.html#suppinfo>; <http://www.nature.com/ncb/journal/v14/n3/index.html#ed>). No restrictions apply to Supplementary Tables or Videos, but we advise authors to be selective in including supplemental data.

GUIDELINES FOR EXPERIMENTAL AND STATISTICAL REPORTING

REPORTING REQUIREMENTS – To improve the quality of methods and statistics reporting in our papers we have recently revised the reporting checklist we introduced in 2013. We are now asking all life sciences authors to complete two items: an Editorial Policy Checklist (found here <https://www.nature.com/authors/policies/Policy.pdf>) that verifies compliance with all required editorial policies and a reporting summary (found here <https://www.nature.com/authors/policies/ReportingSummary.pdf>) that collects information on experimental design and reagents. These documents are available to referees to aid the evaluation of the manuscript. Please note that these forms are dynamic ‘smart pdfs’ and must therefore be downloaded and completed in Adobe Reader. We will then flatten them for ease of use by the reviewers. If you would like to reference the guidance text as you complete the template, please access these flattened versions at <http://www.nature.com/authors/policies/availability.html>.

Author Rebuttal to Initial comments

Reviewers' Comments:

Reviewer #1:

Remarks to the Author:

Coquand et al. describe a developed a method to quantitatively map human bRG cell division modes in the human developing cortex and in cerebral organoids. The method is based on a semi-automated imaging of live-fixed tissue, which enables identification of bRG cell progeny fate following their division. The authors first identify morphology of bRG cells in human embryonic cortical samples and cerebral organoids based on their location basally to VZ areas, expression of SOX2 and pVim, the presence of at least one process and the typical mitotic soma translocation (MST) following cell division. The authors then devise a semiautomated system to follow dividing cells in live tissue followed by fixation, which is based on computerized screening of multiple videos of live cells and matching those to their corresponding fixated and immunostained counterparts (the same cells). This allows the fate analysis of two daughter cells derived from a bRG cell following division.

Based on these analyses, the authors find high rate of proliferative bRG cells (60%, giving rise to 2 bRG cells) in cerebral organoids – as sign for bRG self-renewal in the oSVZ, while a significant portion of bRG cells was consumed for generating neurons by symmetric divisions producing two neurons. This is in addition to the asymmetric division mode producing one bRG cell and one neuron (both modes either directly or via IP cells). To gain more insights on transition through these modes the authors first detect preferential symmetric neurogenic divisions at more distal / basal regions of the oSVZ, while asymmetric self-renewal or symmetrical/asymmetrical neurogenic divisions appeared comparable along the oSVZ. On the other hand, among those asymmetric neurogenic cell divisions (producing bRG and IP cell/neuron) that can be detected more apically in the oSVZ, no correlation between a bRG daughter cell fate and apical or basal process inheritance was found. This was confirmed in both fetal tissue and organoids. To support this, the authors show no preferential Notch signalling (stem cell state) in bRG cells (before or after cell division) that inherited either an apical or a basal process. The authors conclude that a major trajectory for bRG cells consists of proliferative, symmetric self-renewal of bRG cells followed by self-consuming, symmetric divisions that generate neurons directly, independently of IPs. Finally, they conclude that a stem cell fate in daughter cells is irrespective of basal process inheritance. The authors present these findings in light of the sharp contrast to mouse aRG cells that largely rely on IPs to amplify the neurogenic output and on process inheritance for determining stem cell state.

The main and most outstanding finding in this work, which also provides an important novel advancement towards understanding stem cell maintenance and differentiation trajectories in the developing human cortex is that bRG cells expand the human cortex mainly in symmetrical manner. They do so in two major modes that well balance each other in terms of cell pool quantities: the first is the self-expanding proliferative mode where bRG cells symmetrically divide to produce 2 bRG cells; this is followed by the second, self-consuming neurogenic mode, where bRG cells again divide symmetrically, this time to directly produce two neurons. While these modes of cortical expansion are further accompanied by the more conventional trajectories of asymmetrical divisions generating bRG cells and neurons/IP cells or symmetrical divisions producing two IP cells, this study finds evidence that former expansive/consumptive symmetrical mode is dominating the growth of the human cortex during development when compared to the mouse cortex. This is further added to the fact that radial glial processes do not seem to play any role dictating the stem cell stage within the pool of the asymmetrically dividing bRG cells, providing further room for exploring how human bRG cells exert the oSVZ environment to expand the neurogenic pool in an asymmetric manner. This method is highly not trivial and very well appreciated. The technical work is thorough and well planned and done. The choice of GW 14-17 in vivo and week 8-10 cerebral organoids is well matched. And the finding that asymmetrically dividing bRG cells generating one IP and one neuron was never observed is super interesting. Finally, because the data already provide a shift in the way we understand stem cell proliferation and differentiation trajectories in humans, any “natural requests” to take advantage of this method for comparing the findings on bRG cells to aRG cells, to explore the subtypes of neurons (or

even glia) that are produced by these types of bRG modes - all can be simply considered as beyond the scope of this work. Therefore, this work is highly recommended for publication.

We thank the reviewer for their very positive and constructive suggestions that have greatly improved the manuscript. We provide a point-by-point reply to their comments.

There are several points to be considered in any revised manuscript format:

1. bRG cell identity. While the criteria used here to identify bRG cells are thorough, it would be good to get a more direct identity measure such as specific bRG markers. While it is understood that immunostaining for bRG markers such as LIFR1, HOPX and PTPRZ1 may not be conclusive due to the abundance of such markers also in non bRG cells, it would be imperative to show a few cases where co-expression of such markers together in candidate bRG cells is indicated to strengthen the method of bRG cell identification. scRNA-Seq is always the preferred method due the uncertainty of these markers when expressed alone, but again, co-expression at the immunostaining level can do the work. This will also help strengthen the criteria for bRG cell identification in line 140: “double-negative (EOMES-/NEUN-) cells being identified as bRG cells”.

We have now better characterized the identity of these cells, by performing co-staining of SOX2+, EOMES+ and NeuN+ cells with HOPX. We show that virtually all HOPX+ cells are positive for SOX2 and negative for EOMES, NEUN and HuC/D (Extended data figure 1a & b). These results confirm our ability to faithfully identify bRG cells.

2. Line 140: “...We noted that when a daughter cell differentiated (e.g. into an EOMES+ IP), it always retained some expression of the mother cell fate marker (SOX2) (Fig. 2c)”. A deeper look into the figures indeed reveals that SOX2 is retained in the differentiating cell – but only in asymmetrical divisions. Can the authors confirm that this is indeed the case, describe it better in the text, and provide their own view on why symmetrical neurogenic divisions do not show retaining transient SOX2 expression?

We do not find any evidence for differential retention of SOX2 in symmetrically and asymmetrically dividing cells. SOX2 can also be retained in symmetrical self-consuming divisions (Fig. 3a) and be lost in asymmetric divisions (Fig 6a & b). Overall, SOX2 protein appears quite stable in differentiating cells, irrespective of the mode of division that generated it. We do agree that transcription factor post-mitotic stability (and potential function) will be a critical question to address in the future. We now explicit this in the manuscript.

3. Symmetric vs. asymmetric cell division (Line 179). The finding that ~70% of bRG cells divide symmetrically is dramatic for weeks 8-10 organoids (~ day 70). Can the authors test a different (later) time point to show that this may change? In other words, is it a static rate for bRG cells? or does it change with time as occurs for mouse aRG cells?

We agree that how bRG division modes vary in time is an outstanding question, which we now address. First, as requested by reviewers 2 and 3, we now provide data for each individual time point, rather than pooled data covering multiple weeks of development as previously done (Figures 3 and 4). Second, and most importantly, we now add correlative data for several later time points (Weeks 13, 14 and 15 for cerebral organoids and GW 18 for fetal tissue) (Figures 3 and 4). Finally, because we noted that glial cells begin to appear in fetal tissue around GW17-18, we added pan-glial markers (oligodendrocytes + astrocytes) to our analysis (Figure 4).

In cerebral organoids, we observe that self-amplifying (proliferating) divisions increase between weeks 7 and 9 and subsequently decrease between weeks 13 and 15, suggesting a peak around week 11. Within neurogenic divisions, we observe that self-consuming divisions decrease between weeks 7 and 9 and subsequently increase between weeks 13 and 15. Finally, we show that direct neurogenic division decrease between weeks 7 and 9 and subsequently increase between weeks 13 and 15. This indicates that at the stages when bRG cells do more proliferative divisions, their neurogenic divisions are less likely to be asymmetric self-consuming (and more likely to self-consuming) and less likely to be direct. The significance of these complex behaviors is addressed using a mathematical model (see below).

In human fetal tissue, we observe less strong variations between gestational weeks 14 and 18. The most notable phenomena being the decrease of neurogenic divisions in favor of gliogenic divisions (Figure 4e). Note that the time window sampled for fetal tissue is twice smaller than for organoids.

4. Comparison with human fetal tissue: it is not always clear why the authors alternate between fetal and organoid tissue. At some parts, mainly the establishment of the method, it was done in both tissues and was also essential. Other than this, the rationale should be better articulated.

Our manuscript is built as a comparison between fetal tissue and cerebral organoids, which has been under addressed, except for some notable exceptions. One major goal of our approach was to quantify precisely where organoids faithfully recapitulate fetal development, and where may fail to do so.

In figures 1 to 6, we therefore performed a systematic comparison between the two tissues (bRG cell morphologies and fate, cell fate decisions, role of the basal process). We conclude from these experiments that that cell fate decisions are very conserved between organoids and fetal tissue, at least within the boundaries of our experimental approach. We however show that one major difference between the two systems lies in the positional control of cell fate decisions. In fetal tissue, bRG cells divide differently depending on their position in the OSVZ, which does not occur in organoids, simply because their OSVZ is much smaller. Therefore, cortical organoids faithfully recapitulate human fetal neurogenesis at these early stages (abundant bRG amplification, abundant direct divisions and abundant self-consuming divisions), except for the spatial aspects.

Based on this validation, we used cerebral organoids to test for the regulation of asymmetric Notch signaling (Figure 7, note that a functional assay is now added (Extended data Figure 7)). Because Notch activity is highly challenging to monitor in the correlative assay, we focused on the organoid model, where we could obtain much more data in order to recover enough cells. We agree that the logic was not perfectly clear in the text and have reworded this accordingly.

5. A model for division patterns (Fig. 3d). Can the authors calculate a model of cortical expansion from these interesting findings? How many cells are generated that can accommodate for cortical development via these modes compared to before these modes were identified as major in this study (i.e. compare to the use of IP cell rather than direct, and less proliferative symmetrical divisions). In other words, how these modes are advantageous for cortical expansion.

We show here that cell fate decisions in human cortical progenitors are probabilistic behaviors that vary in space and time. As suggested by the reviewer, we have now generated a mathematical model to predict bRG cell output based on the probability of each possible division mode (Figure 3 f, g and extended data Figure a, b).

At week 8, our model shows that, after 4 rounds of divisions, 1 bRG cell leads on average to the generation of 5,75 bRG cells, 1,21 IPs and 2,69 neurons, highlighting their strong self-amplification potential. The final number of neurons is underestimated here, as we do not consider the output of IPs (the probabilities of their outputs remain unknown, as we still cannot unambiguously identify them in live samples). Consistent with the proliferation probabilities, we show that the number a bRG cells generated by a single bRG cell increases from weeks 7 to 9 and decreases from weeks 13 to 15. Strikingly, this occurs at a relatively constant neurogenic rate, indicating that single progenitors produce the same number of differentiated cells, irrespective of their degree of self-amplification. At week 9 for example, bRG cells are less likely to produce a differentiated cell after one division, but this is counterbalanced after several divisions by the increase in their own pool.

Therefore, the production of differentiated cells by single bRG cells after one week of development (4 divisions) is constant at each developmental stage. At the population level however, the production of differentiated cells increases as a consequence of the increased pool of bRG cells.

We furthermore show that a 20% reduction in the rate of symmetrical divisions in favor of asymmetric self-renewing indirect divisions (1bRG + 1 IP) - the dominant division mode in mouse aRG cells at neurogenesis onset – reduced the total production of bRG cells by 31% after only 4 divisions (Extended data Fig. 4b).

Reviewer #2:

Remarks to the Author:

Summary

Coquand et al. develop a novel pipeline for systematically integrating live imaging data with fixed immunohistochemistry and implement it to track the behaviors and lineages of basal progenitors in the developing cortex. This method represents a very useful tool that can increase the throughput and accuracy of studies of a similar nature, and the authors use it to address an important question in the field of cortical development. The authors show in primary tissue and organoids that basal radial glia undergo both proliferative and neurogenic divisions and note a gradient of direct vs. indirect neurogenesis based on proximity to the apical surface of the developing cortex.

While technically impressive, the major findings are not altogether novel and are not very deeply explored. The nature of bRG divisions and the relative frequency of their outputs has been reported in previous studies in human and primate, some of which have gone more in depth by reporting other factors such as mitotic cleavage angles or by tracing progenitors over multiple divisions to more accurately report their proliferative potential. The association with Notch signaling is quite weak in this study, relying on a single IHC panel and the absence of a positive signal to make a claim, whereas previous studies have more thoroughly and directly interrogated the effect of Notch on cell behaviors and daughter fate. The finding of direct vs. indirect neurogenesis correlating with distance from the apical surface does seem novel and is of significant interest to the field, but there is no further investigation of this phenomenon or mechanistic hypotheses.

Overall, this study presents a highly useful method that could benefit the field and show that its use corroborates an established base of knowledge about basal radial glia dynamics. However, the paper currently falls short of significantly expanding knowledge of these cellular behaviors and offers little mechanistic insight. I would invite the authors to more critically analyze the data that they have and follow up their findings with more direct perturbations in order to set their work above the current standard of the field.

We thank the reviewer for their comments and suggestions. We have now performed novel experiments and analyses in order to more thoroughly investigate cell fate decision control during human cortical development.

General comments

- GW14-17 is a broad range of development, particularly with regards to the emergence and abundance of bRG. Samples spanning these ages should not be directly pooled, or data from individual samples should be shown to allow readers to see sample variation across timepoints.

Based on this comment and on reviewer 1 request to analyze later time points, we now present organoid and fetal data as individual time points, ranging from week 7 to 15 (organoids) and GW 14 to 18 (fetal tissue). We hereunder copy the description of this analysis from the Reviewer 1 comment section:

“In cerebral organoids, we observe that self-amplifying (proliferating) divisions increase between weeks 7 and 9 and subsequently decrease between weeks 13 and 15, suggesting a peak around week 11. Within neurogenic divisions, we observe that self-consuming divisions decrease between weeks 7 and 9 and subsequently increase between weeks 13 and 15. Finally, we show that direct neurogenic division decrease between weeks 7 and 9 and subsequently increase between weeks 13 and 15. This indicates that at the stages when bRG cells do more proliferative divisions, their neurogenic divisions are less likely to be asymmetric self-consuming (and more likely to self-consuming) and less likely to be direct. The significance of these complex behaviors is addressed using a mathematical model (see below).

In human fetal tissue, we observe less strong variations between gestational weeks 14 and 18. The most notable phenomena being the decrease of neurogenic divisions in favor of gliogenic divisions (Figure 4e). Note that the time window sampled for fetal tissue is twice smaller than for organoids.”

- The manuscript makes it clear that there are minority populations of these dividing cells that do not conform to generally assumed rules, such as bRG that do not have processes, bRG that undergo stationary divisions, and IPs that have processes. This makes interpretability of some of the later findings challenging when it comes to assigning fates to a progenitor that underwent a consuming division. For example, Figure 2d shows a stationary division that results in generation of two neurons, while Figure 3b shows an MST division that results in generation of two neurons. The mother cell in 3b is labeled as a bRG which seems correct based on the MST division, but it is impossible to assign the mother cell identity in Figure 2d based on the authors' own observation of bRG that undergo stationary divisions. Similarly, Figure 3a shows a stationary division that yields two IPs; the mother cell is assigned as a bRG, but previous works have suggested that IPs may be able to symmetrically divide to make more IPs (Hansen 2010). This assignment of the mother cell has significant implications for the eventual summary figures when it comes to assigning the division modes and outputs of these cell types. Given the shared behavior of stationary division between bRG and IPs and even the shared existence of processes, it seems that any stationary division that does not result in at least one bRG cannot be confidently assigned to a bRG or IP. The authors state that they analyze bRG divisions that undergo MST, which is reassuring, but this caveat introduces a ~20% error in their summary figures by leaving them unable to count stationary bRG divisions.

The fate of the mother cell is a critical aspect of this type of approach and the precise identification criteria were indeed not clearly enough described in the manuscript. We find that the best criteria to identify bRG cells is their elongated morphology. In fetal tissue, 90% of elongated cells are SOX2+ EOMES- and this goes up to 95% in organoids (figures 1h and 1n). Based on this, the potential error margin is therefore quite low (5-10%). Performing the analysis in such a way enabled us to include bRG cells undergoing stationary division.

Here, we tested whether the mode of translocation (static, apical MST, basal MST) correlated with different cell fate decisions (Extended data figure 5c). We found no evidence of such an effect, with proliferative, neurogenic self-consuming, or neurogenic direct divisions being relatively constant in each of the three bRG cell populations. This result indicates that MST is not critical to identify bRG cells and to monitor their cell fate decisions. Cell elongation on the other hand is, as round cells rarely produce bRG daughters (not shown), suggesting an IP fate. We now present these results and better explain the bRG cell identification method.

- The authors surprisingly seem to ignore one of their major findings of difference between organoids and primary cortex, which is that bRG in the cortical tissue undergo substantially more indirect neurogenesis compared to organoids. Organoid bRG directly generated neurons in 50/70 (71%) self-consuming divisions and in 52/81 (64%) self-renewing divisions, while primary bRG directly generated neurons in only 18/46 (40%) self-consuming divisions and 34/92 (37%) self-renewing divisions. This is quite a striking difference that goes completely underreported and raises serious questions about the claim that primary bRG “closely match” organoid bRG, in addition to obscuring a legitimate question of why organoid bRG are biased towards direct neurogenesis.

Because of our novel analysis, throughout multiple developmental time points, the numbers are now slightly different. However, the two major conclusions remain valid:

- 1/ direct neurogenesis is a frequent event in human bRG cells (both in fetal tissue and organoids), unlike what is known for mouse aRG cells. This finding is largely discussed in the manuscript.
- 2/ direct neurogenesis is higher in organoids (53-76%), as compared to fetal tissue (42%). We agree with the reviewer that this later finding should have been better highlighted and discussed, which we now do. The reason for this difference is not clear but could be due to imperfect match of the organoid and fetal tissue stages or to an inherent caveat of organoids. Organoid protocols still lack several features of the human developing cortex (vasculature...), highlighted by a much smaller OSVZ, which may affect cell fate decisions. We nevertheless note that the major cell fate decision patterns (abundant proliferative divisions, and frequent self-consuming and direct neurogenic divisions) are well conserved in organoids.

- The figures in general are somewhat padded with non-essential information. Figures 2c-e, 3a-c, and 4a-d do not inherently add much more information and are essentially extensions of the methods. Even the stacked barcharts on the right side of Figure 5 offer fairly little new information or synthesis relative to the swarmplots. The figures could be made more concise or introduce more analyses/comparisons rather than repeating n of 1 examples that could be presented in extended data. We have now simplified the figures by moving some of these panels to the supplemental figures, and replaced them by novel analyses (Figures 3, 4, 5 and 7).

- The authors seem to miss an opportunity with this study by limiting their live imaging window to 48 hours. While I understand the technical challenges associated with preserving the samples during the imaging and generating more data to analyze, it feels like a valuable opportunity to further explore the fate potential of individual cells. Tracking multiple divisions could also provide more insight on some of the observations made from the poster – if the same bRG divides close to the apical surface during one division and then farther away for a second division, is its fate potential affected as would be predicted by the current results? These longer term experiments have been conducted in the past (Betizeau 2013) and it would be of significant interest to the field to validate those findings from macaque in a human context.

Longer movies (96 hours instead of 48) would indeed allow to track two consecutive divisions and analyze 4-cell clones. These experiments would however not enable to test whether cell fate decisions change during translocation from one region of the OSVZ to another. Indeed, these cell fate differences are observed over hundreds of micrometers, while bRG cells migrate only a couple dozen micrometers between two divisions. The scales of these two phenomena are therefore not compatible. The other outcome of such experiment is to test whether progenitors are multipotent. Do single progenitors undergo different cell fate decisions or are they committed to one type? While these questions are indeed critical, we believe they represent an entire study on their own. They furthermore require a specific sample preparation and imaging setup, that cannot be established within the revision timeframe.

- The Notch experiments are extremely limited to a single HES1 staining panel and miss an opportunity to show with direct intervention that Notch activation is relevant to daughter cell fate or division mode. Again, previous work has explored this question by treating with DAPT to inhibit Notch, which resulted in a bias towards IP fate at the expense of bRG fate (Hansen 2010). The current study has far too low n of 16 cells to make any conclusion about the necessity of Notch for progenitor fate maintenance. A direct perturbation would increase confidence in the authors' finding and would provide better support for their speculation that Notch signaling and its effect on radial glia is related to the cell soma's microenvironment.

We now better document the contribution of NOTCH signaling to human bRG cell self-renewal. Cerebral organoids were infected with GFP-expressing retroviruses and incubated with the gamma-secretase inhibitor DAPT for 2 days before fixation and cell fate analysis. This treatment led to a strong depletion of the bRG population, confirming the critical role of the Notch pathway for self-renewal. Strikingly, this depletion occurred in favor of IPs, but not of neurons, indicating that, when Notch signaling is artificially blocked, bRG cells commit to indirect, but not to direct, neurogenesis (Extended data Figure 7). This result indicates that indirect neurogenesis is the default differentiation pathway in the absence of any Notch signaling, and that direct neurogenesis must be regulated via another mechanism, likely the Robo-Slit pathway as proposed previously (Cardenas et al, 2018). This novel data, together with our previous result showings that Notch signaling is specifically activated in the self-renewing daughter, confirms the critical role of Notch signaling for RG cell self-renewal.

Specific feedback

- Figure 1b

- o insets for Zone 1 and Zone 3 do not seem to match the low-mag image

The insets do come from the positions indicated in the low mag. The difference is due to the fact that the 3 cells are in very different positions in Z, and can therefore be difficult to see in the full stack. We agree that this can be confusing and now only display the high-resolution images of the cells.

- Figure 1h

- o It's unclear if Sox2 and EOMES are considered to be mutually exclusive. Past work has shown that these markers can be co-expressed (Hansen 2010), as do images from later in the paper (Figure 4d) This was indeed not clear in the previous version. We do find instances of SOX2+ EOMES+ double positive cells. Because our correlative data indicates that SOX2 is quite stable in differentiating cells, we consider these cells as EOMES+ IPs. We have now explained this in the legend.

- Figure 5

- o The stacked bar charts do not offer new insight beyond the swarmplots, and there is no reference to them in the text or explanation of why the seemingly-arbitrary 1mm distance from the apical surface was chosen as a relevant metric to bin cells.

- o We agree with the reviewer and have removed these plots.

- Figure 6f

- o The assignment of the bottom cell as a bRG and not a neuron seems quite subjective based on the NeuN staining perfectly colocalizing with the labeled cell, especially when compared to figure 3d which also shows a cell with bright SOX2 and dim NeuN but is called a neuron.

- o The entire region was very low in neurons (apical) and therefore background appeared on this figure as actual signal. But the pixel intensity was very low and therefore considered as negative. We have reset the contrast to proper levels, as it had wrongly been over pushed here.

- Figure 7g

- o The model is very unclear and does not help understand what the authors' main point is. They seem to suggest that there's Notch everywhere in the OSVZ and would be on in IPs as well, but they show in their data that this is not the case.

- o We have removed this model from the figure.

- Discussion: "Our results indicate that a major trajectory for bRG cells consists of symmetric amplifying divisions, followed by self-consuming divisions that generate neurons directly, independently of IPs."

- o There is no basis for this statement in the paper. With the current methods, there is no evidence of temporal ordering or any one division mode that is "followed by" another -- all of the analyses are performed at one timepoint.

- o This is indeed true and we have more carefully worded this in the text.

- Discussion: "aRG cells rely on IPs to amplify their neurogenic output, as their own amplification is limited by spatial constraints... bRG cells on the other hand are not subject to this physical limitation and can amplify their own pool both radially and tangentially, and thus IPs are less relied upon to increase the neurogenic output."

- o This statement is speculative and should be introduced as such. There is little conceptual merit to this idea either – why does producing an IP somehow take up less space than a neuron?

- o What we mean here is not that IPs take less space than neurons, but that aRG self-amplification is much more physically limited than bRG self-amplification (irrespective of direct and indirect division). Because aRG amplification is limited, we propose that IPs may represent a means to increase output and overcome this limitation. We hypothesize that because bRG amplification is much less physically limited (as they can also expand radially, unlike aRGs), they may rely less on IPs to increase their output. We agree that this is entirely speculative, and have therefore reworded the text accordingly.

Reviewer #3:

Remarks to the Author:

Coquand et al. characterize the proliferation mode of basal radial glia (bRG) in both midgestational primary human cortical slices and cerebral brain organoids. They demonstrate that approximately half of the neurogenic divisions by bRG produce neurons directly without going through an intermediate progenitor cell. Moreover, bRGs at greater distance from the apical surface are more likely to go through direct neurogenic divisions. They also found that inheritance of the basal process does not maintain the fate of daughter cells as bRG.

The manuscript is well prepared, and experiments are well designed/executed. Basal radial glia are an important type of neural stem cell and further characterizing cell behavior and fate will generally contribute to our understanding of cortical expansion in mammals.

I have three comments:

1. My major concern is about Figure 5 which led to an important conclusion of the manuscript but was only done in one sample.

We managed to obtain another fresh human fetal brain sample from a developmental stage close to the first one, where OSVZ size and bRG division modes are therefore comparable. This novel sample clearly reproduced our initial observation, that more direct neurogenic divisions are observed away from the ventricular surface (total of N=527 dividing bRG cells).

2. If direct neurogenesis is more common in the upper region of the OSVZ, one would predict that EOMES+ intermediate progenitor cells would be less abundant there relative to radial glia. Could the authors quantify the proportion of radial glia, intermediate progenitor cells, and neurons with respect to their distance from the apical surface in fetal human slices?

We measured the average distance of bRG cells and of IPs in our samples and found no significant difference (around 800 μm on average). We however point out that measuring IP position is much less precise than monitoring the actual birth, as results can be influenced by a variety of factors (e.g. cell cycle duration, cell fate decision of IPs...). For instance, IPs may have a higher self-implication potential basally than apically. Our results would be compatible with a model where, in apical regions, bRGs do more indirect divisions but generate IPs with reduced proliferation potential (as in mouse aRG cells). In basal regions, bRG cells would do less indirect division, but generate IPs that are more proliferative. Fate decisions of IPs (which cannot be unambiguously identified in live samples with the current method) will be a very important topic for the future.

3. In rodents, radial glia divisions are predominantly symmetric early on, produce neurons directly later, and produce neurons indirectly later still. It would be interesting and important to determine if the results reported here, a shift in division from symmetric to neurogenic could be followed by a predominance of indirect neurogenesis. This could be explored by quantifying division outcomes across the sampled ages in fetal tissue (GW 14-18), and/or in organoids which could be cultured for longer timepoints.

We agree that this is a key question, that was also raised by the other reviewers. I hereunder copy our response to reviewer one on this point:

“First, as requested by reviewers 2 and 3, we now provide data for each individual time point, rather than pooled data covering multiple weeks of development as previously done (Figures 3 and 4). Second, and most importantly, we now add correlative data for several later time points (Weeks 13, 14 and 15 for cerebral organoids and GW 18 for fetal tissue) (Figures 3 and 4). Finally, because we noted that glial cells began to appear in fetal tissue around GW17-18, we added pan-glial markers (oligodendrocytes + astrocytes) to our analysis (Figure 4).

In cerebral organoids, we observe that self-amplifying (proliferating) divisions increase between weeks 7 and 9 and subsequently decrease between weeks 13 and 15, suggesting a peak around week 11. Within neurogenic divisions, we observe that self-consuming divisions decrease between weeks 7

and 9 and subsequently increase between weeks 13 and 15. Finally, we show that direct neurogenic division decrease between weeks 7 and 9 and subsequently increase between weeks 13 and 15. This indicates that at the stages when bRG cells do more proliferative divisions, their neurogenic divisions are less likely to be asymmetric self-consuming (and more likely to self-consuming) and less likely to be direct. The significance of these complex behaviors is addressed using a mathematical model (see below).

In human fetal tissue, we observe less strong variations between gestational weeks 14 and 18. The most notable phenomena being the decrease of neurogenic divisions in favor of gliogenic divisions (Figure 4e). Note that the time window sampled for fetal tissue is twice smaller than for organoids.”

Minor comment:

We are told the human fetal tissue came from autopsies. The authors should clarify whether these were normotypic or pathologic samples.

The fetal samples used in this study came either from miscarriages or abortions due to kidney malformations. No brain defects were ever reported. We have clarified this in the text.

Decision Letter, first revision:

Dear Alex,

Your manuscript, "A cell fate decision map reveals abundant direct neurogenesis in the human developing neocortex", has now been seen by all of our original referees, who are experts in cerebral organoids, tissue development, human neocortex, live imaging (referee 1); cerebral organoids, cortex development (referee 2); and brain organoids, human cortex development, radial glial cells, live imaging (referee 3). As you will see from their comments (attached below) they find this work of interest and improved, but have raised some important points. Although we are also very interested in this study, we believe that their concerns should be addressed before we can consider publication in Nature Cell Biology.

We therefore request that you please address all the remaining reviewer points. We are committed to providing a fair and constructive peer-review process, so please feel free to contact me if you would like to discuss any of the referee comments further.

Please pay close attention to our guidelines on statistical and methodological reporting (listed below) as failure to do so may delay the reconsideration of the revised manuscript. In particular please provide:

- a Supplementary Figure including unprocessed images of all gels/blots in the form of a multi-page pdf file. Please ensure that blots/gels are labeled and the sections presented in the figures are clearly indicated.
- a Supplementary Table including all numerical source data in Excel format, with data for different figures provided as different sheets within a single Excel file. The file should include source data giving rise to graphical representations and statistical descriptions in the paper and for all instances where the figures present representative experiments of multiple independent repeats, the source data of all repeats should be provided.

We therefore invite you to take these points into account when revising the manuscript. In addition, when preparing the revision please:

- ensure that it conforms to our format instructions and publication policies (see below and www.nature.com/nature/authors/).
- provide a point-by-point rebuttal to the full referee reports verbatim, as provided at the end of this letter.
- provide the completed Editorial Policy Checklist (found here <https://www.nature.com/authors/policies/Policy.pdf>), and Reporting Summary (found here <https://www.nature.com/authors/policies/ReportingSummary.pdf>). This is essential for reconsideration of the manuscript and these documents will be available to editors and referees in the

event of peer review. For more information see <http://www.nature.com/authors/policies/availability.html> or contact me.

Nature Cell Biology is committed to improving transparency in authorship. As part of our efforts in this direction, we are now requesting that all authors identified as 'corresponding author' on published papers create and link their Open Researcher and Contributor Identifier (ORCID) with their account on the Manuscript Tracking System (MTS), prior to acceptance. ORCID helps the scientific community achieve unambiguous attribution of all scholarly contributions. You can create and link your ORCID from the home page of the MTS by clicking on 'Modify my Springer Nature account'. For more information please visit www.springernature.com/orcid.

[Redacted]

We would like to receive the revision within four weeks. If submitted within this time period, reconsideration of the revised manuscript will not be affected by related studies published elsewhere, or accepted for publication in Nature Cell Biology in the meantime. We would be happy to consider a revision even after this timeframe, but in that case we will consider the published literature at the time of resubmission when assessing the file.

We hope that you will find our referees' comments, and editorial guidance helpful. Please do not hesitate to contact me if there is anything you would like to discuss.

Best wishes,

Stelios

Stylios Lefkopoulos, PhD
He/him/his
Senior Editor
Nature Cell Biology
Springer Nature
Heidelberger Platz 3, 14197 Berlin, Germany

E-mail: stylios.lefkopoulos@springernature.com
Twitter: @s_lefkopoulos

Reviewers' Comments:

Reviewer #1:

Remarks to the Author:

The authors have adequately addressed all issues raised and improved both the clarity of and evidence supporting their claims.

There is one potential misinterpretation of the request in the first comment regarding bRG cell identity. Indeed, the authors have nicely shown SOX2/HOPX cells that are not negative for HuC and EOMES and hence suspected as bRG cells as further corroborated by cell shape and additional cellular properties. However, many VZ (aRG) cells as well are HOPX/SOX2 positive. The authors were therefore asked to provide evidence for multiple bRG markers that are co expressed together with SOX2 as an alternative for scRNA-Seq ("... it would be imperative to show a few cases where co-expression of such markers together in candidate bRG cells."). Can the authors show at least two of the following, such as SOX2/HOPX/PTPRZ1; SOX2/HOPX/LIFR; SOX2/PTPRZ1/LIFR?

Reviewer #2:**Remarks to the Author:**

The authors have addressed all of my prior concerns and the manuscript should be published without delay.

I congratulate the authors on this beautiful study.

Reviewer #3:**Remarks to the Author:**

The authors should be aware that for experiments in fetal brain slices such as the GFP-expressing retrovirus used to label bRG cells, the SOX2+/EOMES-/NEUN- cells (fig 1e-h) could include OPC progenitor cells that are present at the ages they examined. This caveat should be mentioned in the text.

Minor comments that the authors may consider addressing:

- Figure 1.h and n: Although the authors added in the figure legend that SOX2 = SOX2+/EOMES- and EOMES+ = EOMES+(with or without SOX2). It will be more intuitive if the authors can label this information in the figure.
- Figure 3.b: Although the author explained what proliferative vs neurogenic divisions are, it would be nice if the author can make it more clear in the figure. For example, a box surrounding two blue boxes labeled as neurogenic divisions might help.
- Figure 3.c, d, e: Splitting based on time points indeed provided more information. I wonder for each time point, if the author can have data points representing individual organoids being imaged (or an error bar for each timepoint?) instead of averaging all the samples assessed at one time point. This will allow readers to see how consistent/variable this finding is across different samples/individual organoids at each time point.

- Line 255: "Quantification revealed a depletion of GFP+ bRG cells in favor of IPs, but not of neurons". I understand what the author intend to say but it reads a bit confusing. The authors may want to rephrase the sentence to make it more clear regarding the increase and decrease of these cell types.

GUIDELINES FOR SUBMISSION OF NATURE CELL BIOLOGY ARTICLES

ARTICLE FORMAT

ABSTRACT – should not exceed 150 words and should be unreferenced. This paragraph is the most visible part of the paper and should briefly outline the background and rationale for the work, and accurately summarize the main results and conclusions. Key genes, proteins and organisms should be specified to ensure discoverability of the paper in online searches.

TEXT – the main text consists of the Introduction, Results, and Discussion sections and must not exceed 3500 words including the abstract. The Introduction should expand on the background relating to the work. The Results should be divided in subsections with subheadings, and should provide a concise and accurate description of the experimental findings. The Discussion should expand on the findings and their implications. All relevant primary literature should be cited, in particular when discussing the background and specific findings.

REFERENCES – are limited to a total of 70 in the main text and Methods combined,. They must be numbered sequentially as they appear in the main text, tables and figure legends and Methods and must follow the precise style of Nature Cell Biology references. References only cited in the Methods should be numbered consecutively following the last reference cited in the main text. References only associated with Supplementary Information (e.g. in supplementary legends) do not count toward the total reference limit and do not need to be cited in numerical continuity with references in the main text. Only published papers can be cited, and each publication cited should be included in the numbered reference list, which should include the manuscript titles. Footnotes are not permitted.

Methods should be written concisely, but should contain all elements necessary to allow interpretation and replication of the results. As a guideline, Methods sections typically do not exceed 3,000 words. The Methods should be divided into subsections listing reagents and techniques. When citing previous methods, accurate references should be provided and any alterations should be noted. Information must be provided about: antibody dilutions, company names, catalogue numbers and clone numbers for monoclonal antibodies; sequences of RNAi and cDNA probes/primers or company names and catalogue numbers if reagents are commercial; cell line names, sources and information on cell line identity and authentication. Animal studies and experiments involving human subjects must be reported in detail, identifying the committees approving the protocols. For studies involving human subjects/samples, a statement must be included confirming that informed consent was obtained. Statistical analyses and information on the reproducibility of experimental results should be provided in a section titled "Statistics and Reproducibility".

All Nature Cell Biology manuscripts submitted on or after March 21 2016, must include a Data availability statement as a separate section after Methods but before references, under the heading "Data Availability". For Springer Nature policies on data availability see <http://www.nature.com/authors/policies/availability.html>; for more information on this particular policy see <http://www.nature.com/authors/policies/data/data-availability-statements-data-citations.pdf>. The Data availability statement should include:

- Accession codes for primary datasets (generated during the study under consideration and designated as "primary accessions") and secondary datasets (published datasets reanalysed during

the study under consideration, designated as "referenced accessions"). For primary accessions data should be made public to coincide with publication of the manuscript. A list of data types for which submission to community-endorsed public repositories is mandated (including sequence, structure, microarray, deep sequencing data) can be found here <http://www.nature.com/authors/policies/availability.html#data>.

- Unique identifiers (accession codes, DOIs or other unique persistent identifier) and hyperlinks for datasets deposited in an approved repository, but for which data deposition is not mandated (see here for details <http://www.nature.com/sdata/data-policies/repositories>).
- At a minimum, please include a statement confirming that all relevant data are available from the authors, and/or are included with the manuscript (e.g. as source data or supplementary information), listing which data are included (e.g. by figure panels and data types) and mentioning any restrictions on availability.
- If a dataset has a Digital Object Identifier (DOI) as its unique identifier, we strongly encourage including this in the Reference list and citing the dataset in the Methods.

We recommend that you upload the step-by-step protocols used in this manuscript to the Protocol Exchange. More details can be found at www.nature.com/protocolexchange/about.

DISPLAY ITEMS – main display items are limited to 6-8 main figures and/or main tables. For Supplementary Information see below.

FIGURES – Colour figure publication costs \$395 per colour figure. All panels of a multi-panel figure must be logically connected and arranged as they would appear in the final version. Unnecessary figures and figure panels should be avoided (e.g. data presented in small tables could be stated briefly in the text instead).

All imaging data should be accompanied by scale bars, which should be defined in the legend. Cropped images of gels/blots are acceptable, but need to be accompanied by size markers, and to retain visible background signal within the linear range (i.e. should not be saturated). The boundaries of panels with low background have to be demarked with black lines. Splicing of panels should only be considered if unavoidable, and must be clearly marked on the figure, and noted in the legend with a statement on whether the samples were obtained and processed simultaneously. Quantitative comparisons between samples on different gels/blots are discouraged; if this is unavoidable, it has to be performed for samples derived from the same experiment with gels/blots were processed in parallel, which needs to be stated in the legend.

Figures should be provided at approximately the size that they are to be printed at (single column is 86 mm, double column is 170 mm) and should not exceed an A4 page (8.5 x 11"). Reduction to the scale that will be used on the page is not necessary, but multi-panel figures should be sized so that the whole figure can be reduced by the same amount at the smallest size at which essential details in each panel are visible. In the interest of our colour-blind readers we ask that you avoid using red and green for contrast in figures. Replacing red with magenta and green with turquoise are two possible colour-safe alternatives. Lines with widths of less than 1 point should be avoided. Sans serif typefaces, such as Helvetica (preferred) or Arial should be used. All text that forms part of a figure should be

rewritable and removable.

Regardless of format, all figures must be vector graphic compatible files, not supplied in a flattened raster/bitmap graphics format, but should be fully editable, allowing us to highlight/copy/paste all text and move individual parts of the figures (i.e. arrows, lines, x and y axes, graphs, tick marks, scale bars etc). The only parts of the figure that should be in pixel raster/bitmap format are photographic images or 3D rendered graphics/complex technical illustrations.

SUPPLEMENTARY INFORMATION – Supplementary information is material directly relevant to the conclusion of a paper, but which cannot be included in the printed version in order to keep the manuscript concise and accessible to the general reader. Supplementary information is an integral part of a Nature Cell Biology publication and should be prepared and presented with as much care as the main display item, but it must not include non-essential data or text, which may be removed at

the editor's discretion. All supplementary material is fully peer-reviewed and published online as part of the HTML version of the manuscript. Supplementary Figures and Supplementary Notes are appended at the end of the main PDF of the published manuscript.

Unprocessed scans of all key data generated through electrophoretic separation techniques need to be presented in a supplementary figure that should be labeled and numbered as the final supplementary figure, and should be mentioned in every relevant figure legend. This figure does not count towards the total number of figures and is the only figure that can be displayed over multiple pages, but should be provided as a single file, in PDF or TIFF format. Data in this figure can be displayed in a relatively informal style, but size markers and the figures panels corresponding to the presented data must be indicated.

The total number of Supplementary Figures (not including the “unprocessed scans” Supplementary Figure) should not exceed the number of main display items (figures and/or tables (see our Guide to Authors and March 2012 editorial <http://www.nature.com/ncb/authors/submit/index.html#suppinfo>; <http://www.nature.com/ncb/journal/v14/n3/index.html#ed>). No restrictions apply to Supplementary Tables or Videos, but we advise authors to be selective in including supplemental data.

GUIDELINES FOR EXPERIMENTAL AND STATISTICAL REPORTING

REPORTING REQUIREMENTS – To improve the quality of methods and statistics reporting in our papers we have recently revised the reporting checklist we introduced in 2013. We are now asking all life sciences authors to complete two items: an Editorial Policy Checklist (found here <https://www.nature.com/authors/policies/Policy.pdf>) that verifies compliance with all required editorial policies and a Reporting Summary (found here <https://www.nature.com/authors/policies/ReportingSummary.pdf>) that collects information on experimental design and reagents. These documents are available to referees to aid the evaluation of the manuscript. Please note that these forms are dynamic ‘smart pdfs’ and must therefore be downloaded and completed in Adobe Reader. We will then flatten them for ease of use by the reviewers. If you would like to reference the guidance text as you complete the template, please access these flattened versions at <http://www.nature.com/authors/policies/availability.html>.

STATISTICS – Wherever statistics have been derived the legend needs to provide the n number (i.e. the sample size used to derive statistics) as a precise value (not a range), and define what this value represents. Error bars need to be defined in the legends (e.g. SD, SEM) together with a measure of centre (e.g. mean, median). Box plots need to be defined in terms of minima, maxima, centre, and

percentiles. Ranges are more appropriate than standard errors for small data sets. Wherever statistical significance has been derived, precise p values need to be provided and the statistical test used needs to be stated in the legend. Statistics such as error bars must not be derived from $n < 3$. For sample sizes of $n < 5$ please plot the individual data points rather than providing bar graphs. Deriving statistics from technical replicate samples, rather than biological replicates is strongly discouraged. Wherever statistical significance has been derived, precise p values need to be provided and the statistical test stated in the legend.

Author Rebuttal, first revision:

Reviewers' Comments:

Reviewer #1:

Remarks to the Author:

The authors have adequately addressed all issues raised and improved both the clarity of and evidence supporting their claims.

There is one potential misinterpretation of the request in the first comment regarding bRG cell identity. Indeed, the authors have nicely shown SOX2/HOPX cells that are not negative for HuC and EOMES and hence suspected as bRG cells as further corroborated by cellshape and additional cellular properties. However, many VZ (aRG) cells as well are HOPX/SOX2 positive. The authors were therefore asked to provide evidence for multiple bRG markers that are co expressed together with SOX2 as an alternative for scRNA-Seq (“... it would be imperative to show a few cases where co-expression of such markers together in candidate bRG cells.”). Can the authors show at least two of the following, such as SOX2/HOPX/PTPRZ1; SOX2/HOPX/LIFR; SOX2/PTPRZ1/LIFR?

We thank the reviewer for their evaluation of the manuscript.

We now provide further evidence for bRG cell fate using these markers (Extended Data Fig. 1c, d). In particular, we now provide examples of HOPX+ and/or SOX2+ cells also positive for LIFR and PTPRZ1.

Reviewer #2:

Remarks to the Author:

The authors have addressed all of my prior concerns and the manuscript should be published without delay.

I congratulate the authors on this beautiful study.

We thank the reviewer for their evaluation of the manuscript.

Reviewer #3:

Remarks to the Author:

The authors should be aware that for experiments in fetal brain slices such as the GFP-expressing retrovirus used to label bRG cells, the SOX2+/EOMES-/NEUN- cells (fig 1e-h) could include OPC progenitor cells that are present at the ages they examined. This caveat should be mentioned in the text.

We thank the reviewer for their evaluation of the manuscript. We indeed find that gliogenesis appears and increases from week 16, which is why we have included gliogenic markers (OLIG2 and S100 \$\beta\$ ) for the correlative analysis at GW 18 (Figure 4) (OPCs are reported to be OLIG2+). Although the proportion of OPCs remains low at the stages investigated here, we do agree with the reviewer that this remains a small caveat for Figure 1. We therefore now mention this in the main text.

Minor comments that the authors may consider addressing:

- Figure1.h and n: Although the authors added in the figure legend that SOX2 = SOX2+/EOMES- and EOMES+ = EOMES+(with or without SOX2). It will be more intuitive if the authors can label this information in the figure.

We have added this information directly on the figure.

- Figure3.b: Although the author explained what proliferative vs neurogenic divisions are, it would be nice if the author can make it more clear in the figure. For example, a box

surrounding two blue boxes labeled as neurogenic divisions might help.

We have now added a black box to label all neurogenic divisions in figures 3 and 4, which indeed adds clarity to identification of the different categories.

- Figure 3.c, d, e: Splitting based on time points indeed provided more information. I wonder for each time point, if the author can have data points representing individual organoids being imaged (or an error bar for each timepoint?) instead of averaging all the samples assessed at one time point. This will allow readers to see how consistent/variable this finding is across different samples/individual organoids at each time point.

We have now represented the data as an average between independent live imaged organoid slices, and therefore provide error bars for each time points.

- Line 255: “Quantification revealed a depletion of GFP+ bRG cells in favor of IPs, but not of neurons”. I understand what the author intend to say but it reads a bit confusing. The authors may want to rephrase the sentence to make it more clear regarding the increase and decrease of these cell types.

We have rephrased this to avoid any potential confusion

“Quantification revealed a depletion of GFP+ bRG cells and a corresponding increase of IPs (Extended Data Fig 7a, b). The neuronal population was not affected in the timeframe of the experiment, indicating that indirect neurogenesis is the default differentiation pathway in the absence of Notch signaling.”

Decision Letter, second revision:

30th November 2023

Dear Alex,

Thank you for submitting your revised manuscript "A cell fate decision map reveals abundant direct neurogenesis in the human developing neocortex" (NCB-A49570B). It has now been seen by the original referees and their comments are below. The reviewers find that the paper has improved in revision, and therefore we'll be happy in principle to publish it in Nature Cell Biology, pending minor revisions to comply with our editorial and formatting guidelines.

If the current version of your manuscript is in a PDF format, please email us a copy of the file in an editable format (Microsoft Word or LaTeX)-- we cannot proceed with PDFs at this stage.

Thank you again for your interest in Nature Cell Biology. Please do not hesitate to contact me if you have any questions.

Best regards,
Stelios

Stylianos Lefkopoulos, PhD
He/him/his
Senior Editor, Nature Cell Biology
Springer Nature
Heidelberger Platz 3, 14197 Berlin, Germany

E-mail: stylianos.lefkopoulos@springernature.com
Twitter: @s_lefkopoulos
LinkedIn: [linkedin.com/in/stylianos-lefkopoulos-81b007a0](https://www.linkedin.com/in/stylianos-lefkopoulos-81b007a0)

Reviewer #1 (Remarks to the Author):

The authors have addressed my last concern.
The manuscript can and should be published with no further delay.

Reviewer #3 (Remarks to the Author):

The authors have satisfactorily addressed my concerns.

Decision Letter, final checks:

Our ref: NCB-A49570B

13th December 2023

Dear Dr. Baffet,

Thank you for your patience as we've prepared the guidelines for final submission of your Nature Cell Biology manuscript, "A cell fate decision map reveals abundant direct neurogenesis in the human developing neocortex" (NCB-A49570B). Please carefully follow the step-by-step instructions provided in the attached file, and add a response in each row of the table to indicate the changes that you have made. Please also check and comment on any additional marked-up edits we have proposed within the text. Ensuring that each point is addressed will help to ensure that your revised manuscript can be swiftly handed over to our production team.

In recognition of the time and expertise our reviewers provide to Nature Cell Biology's editorial process, we would like to formally acknowledge their contribution to the external peer review of your manuscript entitled "A cell fate decision map reveals abundant direct neurogenesis in the human developing neocortex". For those reviewers who give their assent, we will be publishing their names alongside the published article.

Nature Cell Biology offers a Transparent Peer Review option for new original research manuscripts submitted after December 1st, 2019. As part of this initiative, we encourage our authors to support increased transparency into the peer review process by agreeing to have the reviewer comments, author rebuttal letters, and editorial decision letters published as a Supplementary item. When you submit your final files please clearly state in your cover letter whether or not you would like to participate in this initiative. Please note that failure to state your preference will result in delays in accepting your manuscript for publication.

Cover suggestions

COVER ARTWORK: We welcome submissions of artwork for consideration for our cover. For more information, please see our guide for cover artwork.

Nature Cell Biology has now transitioned to a unified Rights Collection system which will allow our Author Services team to quickly and easily collect the rights and permissions required to publish your work. Approximately 10 days after your paper is formally accepted, you will receive an email in providing you with a link to complete the grant of rights. If your paper is eligible for Open Access, our Author Services team will also be in touch regarding any additional information that may be required to arrange payment for your article.

Please note that *Nature Cell Biology* is a Transformative Journal (TJ). Authors may publish their research with us through the traditional subscription access route or make their paper immediately open access through payment of an article-processing charge (APC). Authors will not be required to make a final decision about access to their article until it has been accepted. Find out more about Transformative Journals

Please use the following link for uploading these materials:
[Redacted]

Best regards,

Adam Lipkin
Staff
Nature Cell Biology

On behalf of

Stylianos Lefkopoulos, PhD
He/him/his

Senior Editor, Nature Cell Biology
Springer Nature
Heidelberger Platz 3, 14197 Berlin, Germany

E-mail: stylianos.lefkopoulos@springernature.com
Twitter: @s_lefkopoulos
LinkedIn: [linkedin.com/in/stylianos-lefkopoulos-81b007a0](https://www.linkedin.com/in/stylianos-lefkopoulos-81b007a0)

Reviewer #1:

Remarks to the Author:

The authors have addressed my last concern.

The manuscript can and should be published with no further delay.

Reviewer #3:

Remarks to the Author:

The authors have satisfactorily addressed my concerns.

Final Decision Letter:

Dear Alex,

I am pleased to inform you that your manuscript, "A cell fate decision map reveals abundant direct neurogenesis bypassing intermediate progenitors in the human developing neocortex", has now been accepted for publication in Nature Cell Biology. Congratulations to you and the whole team!

Due to the importance of these deadlines, we ask that you please let us know now whether you will be

difficult to contact over the next month. If this is the case, we ask you provide us with the contact information (email, phone and fax) of someone who will be able to check the proofs on your behalf, and who will be available to address any last-minute problems.

Please note that *Nature Cell Biology* is a Transformative Journal (TJ). Authors may publish their research with us through the traditional subscription access route or make their paper immediately open access through payment of an article-processing charge (APC). Authors will not be required to make a final decision about access to their article until it has been accepted. Find out more about Transformative Journals

If you have not already done so, we strongly recommend that you upload the step-by-step protocols used in this manuscript to the Protocol Exchange (www.nature.com/protocolexchange), an open online resource established by Nature Protocols that allows researchers to share their detailed experimental know-how. All uploaded protocols are made freely available, assigned DOIs for ease of citation and are fully searchable through nature.com. Protocols and Nature Portfolio journal papers in which they are used can be linked to one another, and this link is clearly and prominently visible in the online versions of both papers. Authors who performed the specific experiments can act as primary authors for the Protocol as they will be best placed to share the methodology details, but the Corresponding Author of the present research paper should be included as one of the authors. By uploading your Protocols to Protocol Exchange, you are enabling researchers to more readily reproduce or adapt the methodology you use, as well as increasing the visibility of your protocols and papers. You can also establish a dedicated page to collect your lab Protocols. Further information can be found at www.nature.com/protocolexchange/about

With kind regards,
Stelios

Stylianos Lefkopoulos, PhD
He/him/his
Senior Editor, Nature Cell Biology
Springer Nature
Heidelberger Platz 3, 14197 Berlin, Germany

E-mail: stylianos.lefkopoulos@springernature.com
Twitter: @s_lefkopoulos
LinkedIn: [linkedin.com/in/stylianos-lefkopoulos-81b007a0](https://www.linkedin.com/in/stylianos-lefkopoulos-81b007a0)
